# Towards Scalable Bayesian Transformers: Investigating stochastic subset selection for NLP

**Peter J.T. Kampen**[1,*]          **Gustav R.S. Als**[1,*]          **Michael Riis Andersen**[1]

[1]DTU Compute, Technical University of Denmark, Kongens Lyngby, Denmark
[*]These authors contributed equally,

## Abstract

Bayesian deep learning provides a framework for quantifying uncertainty. However, the scale of modern neural networks applied in Natural Language Processing (NLP) limits the usability of Bayesian methods. Subnetwork inference aims to approximate the posterior by selecting a stochastic parameter subset for inference, thereby allowing scalable posterior approximations. Determining the optimal parameter space for subnetwork inference is far from trivial. In this paper, we study partially stochastic Bayesian neural networks in the context of transformer models for NLP tasks for the Laplace approximation (LA) and Stochastic weight averaging - Gaussian (SWAG). We propose heuristics for selecting which layers to include in the stochastic subset. We show that norm-based selection is promising for small subsets, and random selection is superior for larger subsets. Moreover, we propose Sparse-KFAC (S-KFAC), an extension of KFAC LA, which selects dense stochastic substructures of linear layers based on parameter magnitudes. S-KFAC retains performance while requiring substantially fewer stochastic parameters and, therefore, drastically limits memory footprint.

## 1 INTRODUCTION

The field of Natural Language Processing (NLP) has seen extraordinary advances recently with the introduction of the transformer architecture [Vaswani et al., 2017, Devlin et al., 2019, ope, 2023]. Increased size and variety of data have improved calibration. However, many models still suffer from overconfident predictions Tian et al. [2023], Jiang et al. [2021], He et al. [2023]. Bayesian deep learning tackles uncertainty estimation by modeling neural network parameters as distributions, thereby gaining an inherent interpretation of model confidence. High-dimensional parameter spaces nevertheless limit application on transformer models. The advent of subnetwork inference can potentially improve Bayesian inference in transformer models by estimating the posterior distribution over a subset of the model weights while treating the majority of the weights as deterministic. Hence the dimensionality of the parameter space of the stochastic subnetwork will be significantly reduced, thus increasing the feasibility of applying Bayesian methodology[Daxberger et al., 2022b, Sharma et al., 2023].

Bayesian methods for transformers have been explored with varying results [Tran et al., 2019, Chen and Li, 2023, Cinquin et al., 2021]. Sharma et al. [2023] finds that partial stochastic neural networks (NN) can outperform their fully stochastic versions using feed-forward neural networks for regression and convolutional neural networks for image classification. In this paper, we investigate whether the hypothesis of partial stochasticity improving predictive performance compared to both point estimates and fully stochastic solutions can be extended to the NLP domain. The concept of partial stochasticity raises the question of how to select the stochastic subset. We propose heuristics for selecting which affine transformations to include in the stochastic subset for a transformer encoder model (DistilBERT [Sanh et al., 2020]). The study is performed using two methods for approximating the posterior distribution: Stochastic weight averaging - Gaussian (SWAG) [Maddox et al., 2019] and Laplace approximation (LA) [MacKay, 1992b, Daxberger et al., 2022a]. Both methods rely on a maximum a posteriori (MAP) parameter estimate, where the stochasticity is induced post hoc to the designated parameters. To study the trade-off between performance and memory requirements, we construct *ramping* experiments, where the fraction of stochastic vs. total number of parameters varies, such that the performance between different subsets can be evaluated.

We propose a selection strategy for the Kronecker-Factored Approximate (KFAC) LA, dubbed Sparse-KFAC (S-KFAC). Here, fully connected substructures of all linear layers are included in the stochastic subset based on parameter magni-

tudes. Thus, it aims to capture uncertainty from all components of the model while limiting the number of stochastic parameters. Below, we summarize our contributions:

1. We design and implement a set of numerical experiments to investigate the hypothesis of partial stochasticity improving predictive performance for transformer models in the NLP domain.

2. We propose and evaluate novel heuristics for efficient identification of optimal subsets of stochastic parameters.

3. We demonstrate that the proposed S-KFAC Laplace approximation yields competitive predictive performance for substantially fewer stochastic parameters, leading to a reduced memory footprint.

## 2 RELATED WORKS

**Bayesian Deep Learning Methods.** Bayesian deep learning is a promising approach for uncertainty quantification [Abdar et al., 2021a]. Regrettably, the dimensionality of the parameter space in large neural networks makes exact posterior inference intractable. Several algorithmic approaches for approximating the posterior $p(\boldsymbol{w}|\mathcal{D})$ have been proposed [Magris and Iosifidis, 2023]. Markov Chain Monte Carlo (MCMC) methods are prominent due to their ability to sample from the true posterior and Hamiltonian Monte Carlo (HMC) [Neal, 1996] is the gold-standard posterior inference method by leveraging Hamiltonian dynamics in its sampling strategy, but HMC is prohibitively slow for even moderately sized models [Izmailov et al., 2021]. Variational inference (VI) has been tested extensively [Wang and Yeung, 2016, Swiatkowski et al., 2020, Tran et al., 2019]. It aims to approximate $p(\boldsymbol{w}|\mathcal{D})$ using a fixed distributional family, often chosen as a Mean-Field (MF) Gaussian. However, it is known to exhibit pathological behavior [Foong et al., 2020]. Estimating $p(\boldsymbol{w}|\mathcal{D})$ using the Gaussian functional form is also explored in the LA, where a Gaussian is fitted at the mode of the posterior distribution, using the inverse curvature as a covariance approximation [MacKay, 1992a,b]. The post hoc and predictive performance-preserving nature of the LA makes it readily applicable [Daxberger et al., 2022a]. Methods utilizing Stochastic Gradient Descent (SGD) iterates as samples from the approximate posterior such as Stochastic weight averaging (SWA) [Izmailov et al., 2019] and SWA-Gaussian (SWAG) [Maddox et al., 2019] have also been examined. Much less computationally demanding methods like Monte Carlo (MC) dropout exist, where the dropout mechanism normally used at training time for regularization is extended to inference time [Abdar et al., 2021b]. This, too, has been shown to exhibit pathological behavior [Foong et al., 2020].

**Subnetwork Inference In Neural Networks**. The concept of treating only a subset of the model parameters as stochas-

tic has increased in popularity recently Daxberger et al. [2022b], Sharma et al. [2023]. In Daxberger et al. [2022a], the linearized LA [Immer et al., 2021] is adapted for subnetworks and made readily applicable for neural networks using the generalized Gauss-Newton (GGN) Hessian approximation. The efficacy of the approximate posterior inference over a stochastic subset of neural network parameters has been investigated in Sharma et al. [2023]. They argue that partially stochastic models are no less theoretically founded than fully Bayesian models. Additionally, they show that inference over a stochastic subset can, at times, yield better performance than full-model stochasticity. Finally, injecting auxiliary stochastic variables through node-based methods has been explored in Dusenberry et al. [2020], Trinh et al. [2022]. This achieves partial stochasticity without altering the existing model parameters.

**Bayesian Methods in NLP.** Transformer-based models for NLP have improved rapidly in recent years [Vaswani et al., 2017, Devlin et al., 2019, ope, 2023]. However, after the fine-tuning process, problems such as overconfidence and sub-optimal calibration persist [Tian et al., 2023, Jiang et al., 2021, He et al., 2023]. In Xiao et al. [2022], calibration of pre-trained language models is investigated for methods that imitate uncertainty modeling, such as Temperature scaling [Guo et al., 2017] and MC Dropout [Gal and Ghahramani, 2016]. Subnetwork inference on transformer models has been tested using Gaussian Mean-Field VI in Xue et al. [2021], Cinquin et al. [2021], and through Sparse Gaussian Processes in Chen and Li [2023].in Yang et al. [2023], the Laplace approximation is applied to Low-Rank Adaptions (LoRA) for fine-tuning large language models. Additionally, Last Layer Laplace showed similar performance to temperature scaling when applied to common NLP tasks in Daxberger et al. [2022a]. In Talman et al. [2023] Stochastic Weight Averaging - Gaussian [Maddox et al., 2019] was used to induce uncertainty awareness in Natural Language Inference tasks.

## 3 BACKGROUND

In this paper, we study partially stochastic Bayesian neural networks with a primary focus on supervised classification tasks. A neural network $f_{\boldsymbol{w}} : \mathbb{R}^D \to \mathbb{R}^C$, where $D$ is the input dimension and $C$ is the number of classes, parameterized by $\boldsymbol{w}$ is introduced. The likelihood of the data $\mathcal{D} = \{(\boldsymbol{x}_i, t_i)\}_{i=1}^N$ given the parameters $\boldsymbol{w}$ is then written as $p(\mathcal{D}|\boldsymbol{w}) = \prod_{i=1}^N p(t_i|f_{\boldsymbol{w}}(\boldsymbol{x}_i))$. In Bayesian deep learning, Bayes' theorem relates a prior distribution over the neural network parameters $p(\boldsymbol{w})$ to a posterior $p(\boldsymbol{w}|\mathcal{D})$ through a likelihood $p(\mathcal{D}|\boldsymbol{w})$ and a marginal distribution $p(\mathcal{D}) = \int p(\mathcal{D}|\boldsymbol{w})p(\boldsymbol{w})d\boldsymbol{w}$ over the data, by $p(\boldsymbol{w}|\mathcal{D}) = \frac{p(\mathcal{D}|\boldsymbol{w})p(\boldsymbol{w})}{p(\mathcal{D})}$. In this paper we generally assume a Gaussian prior distribution $p(\boldsymbol{w}) = \mathcal{N}(\boldsymbol{w}|\boldsymbol{\mu}, \boldsymbol{V})$. Predictions are made using the posterior predictive distribution

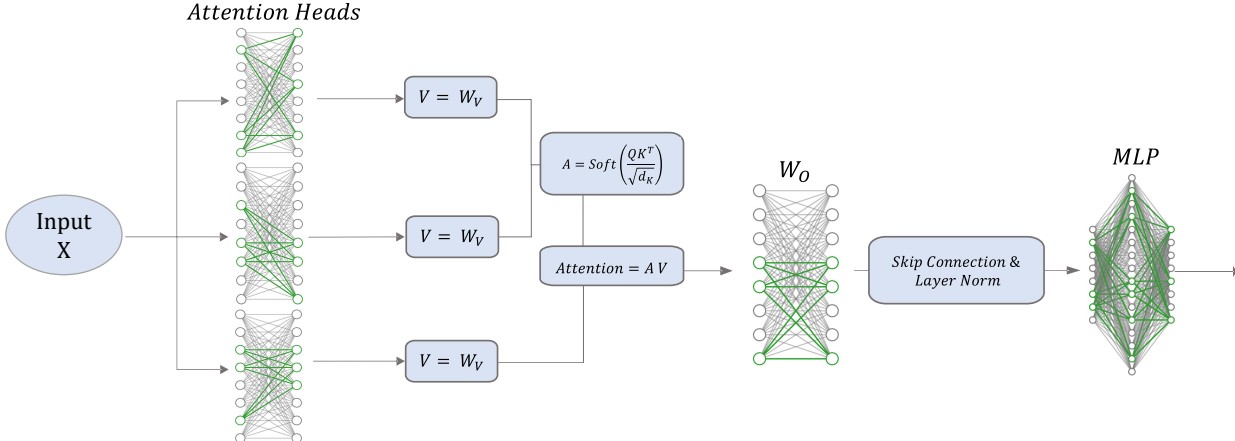

Figure 1: Depiction of a transformer block with stochastic parameters as induced by the Sparse-KFAC LA Section 4. Note that each linear layer is written in neural style, where small dense substructures have been made stochastic, indicated by green lines rather than gray for the deterministic weights. 'Soft' denotes the softmax function.

such that for a new data point $\boldsymbol{x}^*$ the predictive distribution becomes $p(t^*|\boldsymbol{x}^*, \mathcal{D}) = \mathbb{E}_{p(\boldsymbol{w}|\mathcal{D})}[p(t^*|f_{\boldsymbol{w}}(\boldsymbol{x}^*)]$. As the integral of the marginal $p(\mathcal{D})$ is often intractable for neural networks, we have to rely on approximate posterior inference techniques such as SWAG and the LA to perform Bayesian inference. In both methods, we approximate the posterior distribution $p(\boldsymbol{w}|\mathcal{D})$ with a Gaussian distribution $q(\boldsymbol{w}) = \mathcal{N}(\boldsymbol{w}|\boldsymbol{\mu}, \boldsymbol{\Sigma})$ such that $q(\boldsymbol{w}) \approx p(\boldsymbol{w}|\mathcal{D})$.

We define a partially stochastic Bayesian neural network $f_{\boldsymbol{w}}$ parameterized by $\boldsymbol{w} = \boldsymbol{w}_D \cup \boldsymbol{w}_S$. Here $\boldsymbol{w}_D$ and $\boldsymbol{w}_S$ are disjoint sets encompassing all the model parameters, where $\boldsymbol{w}_D$ are the deterministic parameters (point estimates) and $\boldsymbol{w}_S$ are the stochastic parameters (distributional). Hence, the partially stochastic posterior is approximated as $p(\boldsymbol{w}|\mathcal{D}) \approx p(\boldsymbol{w}_S|\mathcal{D}) \prod_d \delta(\hat{\boldsymbol{w}} - \boldsymbol{w}_d) \approx q(\boldsymbol{w}_S) \prod_d \delta(\hat{\boldsymbol{w}} - \boldsymbol{w}_d)$, where $\delta$ is the Dirac's delta distribution. The following sections deal with approximating the posterior distribution of the stochastic weights $p(\boldsymbol{w}_S) \approx q(\boldsymbol{w}_S)$. For readability, we will not use the subset notation, and hence, $p(\boldsymbol{w})$ will refer to the distribution of the parameters for the stochastic subnetwork unless otherwise stated.

### 3.1 STOCHASTIC WEIGHT AVERAGING - GAUSSIAN (SWAG)

Maddox et al. [2019] introduces SWAG as a method for Bayesian inference using SGD to obtain a Gaussian approximation of the posterior distribution. The first two moments are estimated using running averages of the first and second uncentered moments using Welford's algorithm [Welford, 1962] seen in Eq. (1).

$$\overline{\boldsymbol{w}}_{n+1} = \frac{n\overline{\boldsymbol{w}}_n + \boldsymbol{w}_n}{n+1}, \qquad \overline{\boldsymbol{w}^2}_{n+1} = \frac{n\overline{\boldsymbol{w}^2}_n + \boldsymbol{w}_n^2}{n+1} \quad (1)$$

where $\overline{\boldsymbol{w}}_n$ and $\overline{\boldsymbol{w}^2}_n$ are the first and second running mo-

ments, respectively, and the square is taken elementwise. The updates are performed $N$ times over the cause of $M$ SGD iterations for $N < M$. For every iteration, we also save the next column in the deviation matrix given by $\boldsymbol{w}_i - \overline{\boldsymbol{w}}_i$, where the $k$ most recent columns are kept. The moment estimates are denoted as $\boldsymbol{w}_{swa} = \overline{\boldsymbol{w}}_N$, and a diagonal matrix is defined by the second moments as $\boldsymbol{\Sigma}_{diag} = \overline{\boldsymbol{w}}_N^2 - \overline{\boldsymbol{w}^2}_N$. In Eq. (2) a low-rank approximation is done with the $k$ columns of the deviation matrix $\boldsymbol{D}$

$$\text{Cov}(\boldsymbol{w}) = \frac{1}{N-1} \sum_{i=1}^N (\boldsymbol{w}_i - \boldsymbol{w}_{swa})(\boldsymbol{w}_i - \boldsymbol{w}_{swa})^T \quad (2)$$

$$\approx \frac{1}{k-1} \sum_{i=1}^k (\boldsymbol{w}_i - \overline{\boldsymbol{w}}_i)(\boldsymbol{w}_i - \overline{\boldsymbol{w}}_i)^T = \frac{1}{k-1} \boldsymbol{D}\boldsymbol{D}^T. \quad (3)$$

The covariance matrix is then estimated using a combination of the low-rank approximation and $\boldsymbol{\Sigma}_{diag}$, yielding the following posterior approximation for the weights $\boldsymbol{w}$

$$p(\boldsymbol{w} \mid \mathcal{D}) \approx \mathcal{N}\left(\boldsymbol{w}|\boldsymbol{w}_{swa}, \frac{1}{2}\boldsymbol{\Sigma}_{diag} + \frac{1}{2(k-1)}\boldsymbol{D}\boldsymbol{D}^T\right). \quad (4)$$

Finally, to sample from the approximate posterior, the following identity is used [Maddox et al., 2019] $\tilde{\boldsymbol{w}} = \boldsymbol{w}_{swa} + \frac{1}{\sqrt{2}}\boldsymbol{\Sigma}_{diag}^{\frac{1}{2}}\boldsymbol{z}_1 + \frac{1}{\sqrt{2(k-1)}}\boldsymbol{D}\boldsymbol{z}_2$ where $\boldsymbol{z}_1 \sim \mathcal{N}(0, \boldsymbol{I}_d)$, $\boldsymbol{z}_2 \sim \mathcal{N}(0, \boldsymbol{I}_k)$, and here, the $d$ denotes the number of stochastic parameters in the model.

### 3.2 THE LAPLACE APPROXIMATION

The Laplace approximation (LA) MacKay [1992a,b] approximates the posterior $p(\boldsymbol{w}|\mathcal{D})$ using a Gaussian distribution.

$$p(\boldsymbol{w}|\mathcal{D}) = \frac{p(\mathcal{D}|\boldsymbol{w})p(\boldsymbol{w})}{p(\mathcal{D})} = \frac{1}{Z}f(\boldsymbol{w}) \approx \mathcal{N}(\boldsymbol{w}|\boldsymbol{\mu}, \boldsymbol{\Sigma}), \quad (5)$$

where $f(\boldsymbol{w})$ is the unnormalized posterior, $\boldsymbol{\mu}$ is the mean and $\boldsymbol{\Sigma}$ is the covariance, and $Z = \int p(\mathcal{D}|\boldsymbol{w})p(\boldsymbol{w})d\boldsymbol{w}$ defines the normalizing constant. We assume a Gaussian prior $p(\boldsymbol{w}) = \mathcal{N}(\boldsymbol{0}, \sigma^2\boldsymbol{I})$ and apply a second-order Taylor expansion around the MAP solution in log-space

$$\ln f(\boldsymbol{w}) \approx \ln f(\boldsymbol{w}_{\text{MAP}}) - \frac{1}{2}(\boldsymbol{w} - \boldsymbol{w}_{\text{MAP}})^T \mathcal{H}_{\boldsymbol{w}}(\boldsymbol{w} - \boldsymbol{w}_{\text{MAP}}), \quad (6)$$

where $\mathcal{H}_{\boldsymbol{w}} = -\nabla^2_{\boldsymbol{w}}\ln f(\boldsymbol{w})|_{\boldsymbol{w}_{\text{MAP}}}$ is the Hessian estimated at the mode. It can be described in terms of the contributions from the likelihood and prior by

$$\mathcal{H}_{\boldsymbol{w}} = \frac{1}{\sigma^2}\boldsymbol{I} - \sum_{i=1}^{N} \nabla^2_{\boldsymbol{w}} \ln p(t_i|f_{\boldsymbol{w}}(\boldsymbol{x}_i))|_{\boldsymbol{w}_{\text{MAP}}} \quad (7)$$

where $\sigma^2$ is the prior variance. Finally, the Gaussian posterior approximation can be written as

$$p(\boldsymbol{w}|\mathcal{D}) \approx q(\boldsymbol{w}) = \mathcal{N}(\boldsymbol{w}|\boldsymbol{w}_{\text{MAP}}, \mathcal{H}_{\boldsymbol{w}}^{-1}). \quad (8)$$

**The Linearized LA** The Hessian $\mathcal{H}_{\boldsymbol{w}}$ is computationally demanding to calculate for large models and datasets. The generalized Gauss-Newton (GGN) [Schraudolph, 2002, Martens, 2020] approximates the Hessian using first-order derivatives on the parameters and second-order derivatives on the outputs of the model

$$\mathcal{G} \equiv \sum_{n=1}^{N} \mathcal{J}_{\boldsymbol{w}}(\boldsymbol{x}_n)^T \nabla^2_{f_{\boldsymbol{w}}} \ln p(t_n|f_{\boldsymbol{w}_M}(\boldsymbol{x}_n)) \mathcal{J}_{\boldsymbol{w}}(\boldsymbol{x}_n). \quad (9)$$

Here $f_{\boldsymbol{w}_M} = f_{\boldsymbol{w}_{\text{MAP}}}$. The covariance of the LA becomes $\boldsymbol{\Sigma} = \mathcal{H}_w^{-1} \approx \left(-\mathcal{G} + \frac{1}{\sigma^2}\boldsymbol{I}\right)^{-1}$. The GGN also ensures positive semi-definiteness . Estimating the posterior predictive distribution using MC sampling can be sub-optimal [Foong et al., 2019]. In Immer et al. [2021], a local linearization of the LA predictive is recommended. A first-order Taylor expansion is done around the model outputs

$$f_{\boldsymbol{w}}(\boldsymbol{x}) \approx f_{\boldsymbol{w}_{\text{MAP}}}(\boldsymbol{x}) + \mathcal{J}_{\boldsymbol{w}_{\text{MAP}}}(\boldsymbol{w} - \boldsymbol{w}_{\text{MAP}}), \quad (10)$$

here $\mathcal{J}_{\boldsymbol{w}_{\text{MAP}}}(\boldsymbol{x}^*) = \nabla_{\boldsymbol{w}} f_{\boldsymbol{w}}(\boldsymbol{x}^*)|_{\boldsymbol{w}_{\text{MAP}}}$ is the Jacobian of the outputs at the MAP solution. The predictive covariance is then defined by $\boldsymbol{\Sigma}^* = \mathcal{J}_{\boldsymbol{w}_{\text{MAP}}}^T(\boldsymbol{x}^*)\boldsymbol{\Sigma}\mathcal{J}_{\boldsymbol{w}_{\text{MAP}}}(\boldsymbol{x}^*)$ for an input $\boldsymbol{x}^*$. The posterior predictive for classification can be expressed using the Softmax as the inverse link function

$$p(t^*|\boldsymbol{x}^*, \mathcal{D}) = \int_{R^C} \text{Softmax}\left(f(\boldsymbol{x}^*)\right)$$
$$\mathcal{N}\left(f(\boldsymbol{x}^*)|f_{\boldsymbol{w}_{\text{MAP}}}(\boldsymbol{x}^*), \boldsymbol{\Sigma}^*\right) df(\boldsymbol{x}^*), \quad (11)$$

where $C$ is the number of classes and $f(\boldsymbol{x}^*)$ is the marginal distribution over the neural network's outputs. Following Daxberger et al. [2022a], the extended probit approximation is applied such that predictive distribution becomes

$$p(t^* = c|\boldsymbol{x}^*, \mathcal{D}) = \frac{\exp\left(f_{\boldsymbol{w}_M}(\boldsymbol{x}^*)_c T_c\right)}{\sum_{i=1}^{C} \exp\left(f_{\boldsymbol{w}_M}(\boldsymbol{x}^*)_i T_i\right)}, \quad (12)$$
$$T_i = \left(1 + \frac{\pi}{8}\boldsymbol{\Sigma}^*_{i,i}\right)^{-\frac{1}{2}}.$$

Note here that only the diagonal covariance structure is considered in the distribution of the outputs.

**Kronecker Factored Approximate Curvature (KFAC).** Although the GGN limits the computational requirements through linearization, both it and the Hessian still scale quadratically in the number of parameters, $\mathcal{O}(|\boldsymbol{w}|^2)$. To combat this issue, the block-diagonal factorization KFAC [Martens and Grosse, 2020] is used to approximate the GGN. This factorizes the empirical Fisher information matrix (EFIM) $\mathcal{F}$ of each layer independently as the product of two smaller matrices. Note that the EFIM is equivalent to the GGN under common likelihoods [Martens, 2020]. The full EFIM over all parameters is given by

$$\mathcal{F} = \mathbb{E}_{p(\mathcal{D})}\left[\nabla_{\boldsymbol{w}} \ln p(\mathcal{D}|\boldsymbol{w})\nabla_{\boldsymbol{w}} \ln p(\mathcal{D}|\boldsymbol{w})^T\right]. \quad (13)$$

In the KFAC approximation, layers are assumed to be independent, giving rise to the block-diagonal structure. The Fisher block for the $i$-th layer can be written as

$$\mathcal{F}_i = \mathbb{E}_{p(\mathcal{D})}\left[\nabla_{\boldsymbol{w}^{(i)}} \ln p(\mathcal{D}|\boldsymbol{w})\nabla_{\boldsymbol{w}^{(i)}} \ln p(\mathcal{D}|\boldsymbol{w})^T\right]. \quad (14)$$

Let $l_i$ be the $i$'th layer and define $\hat{\boldsymbol{x}}$ as the input to that layer then $l_i(\hat{\boldsymbol{x}}) = \boldsymbol{W}\hat{\boldsymbol{x}} = \boldsymbol{a}$, where $\text{vec}(\boldsymbol{W}) = \boldsymbol{w}^{(i)}$ and bias is omitted for clarity. The vec operator denotes the vectorization of a matrix, independent of whether it is column- or row-major, as long as there is consistency throughout. $\boldsymbol{W} \in \mathbb{R}^{n \times m}$ is the weight matrix, and $\boldsymbol{a}$ is the pre-activation. The gradient is then defined as $\boldsymbol{\delta} = \frac{\partial \ln p(\mathcal{D}|\boldsymbol{w})}{\partial \boldsymbol{a}}$, and the chain rule gives for an input $\boldsymbol{x}$ with target $t$

$$\frac{\partial \ln p(\mathcal{D}|\boldsymbol{w})}{\partial \boldsymbol{W}} = \nabla_{\boldsymbol{w}^{(i)}} \ln p(t|\boldsymbol{x}, \boldsymbol{w}) = \text{vec}(\hat{\boldsymbol{x}}\boldsymbol{\delta}^T) = \hat{\boldsymbol{x}} \otimes \boldsymbol{\delta}. \quad (15)$$

where $\otimes$ is the Kronecker product and $\hat{\boldsymbol{x}}$ denotes the input to the specific layer for an input $\boldsymbol{x}$ to the model. The KFAC approximation of the Fisher block can then be expressed as

$$\mathcal{F}_i = \mathbb{E}_{p(\mathcal{D})}\left[\nabla_{\boldsymbol{w}^{(i)}} \ln p(\mathcal{D}|\boldsymbol{w})\nabla_{\boldsymbol{w}^{(i)}} \ln p(\mathcal{D}|\boldsymbol{w})^T\right]$$
$$= \mathbb{E}[\hat{\boldsymbol{x}}\hat{\boldsymbol{x}}^T \otimes \boldsymbol{\delta}\boldsymbol{\delta}^T] \approx \mathbb{E}[\hat{\boldsymbol{x}}\hat{\boldsymbol{x}}^T] \otimes \mathbb{E}[\boldsymbol{\delta}\boldsymbol{\delta}^T] \quad (16)$$
$$= \boldsymbol{A} \otimes \boldsymbol{G}.$$

Here, the expectation of the Kronecker product is approximated by the Kronecker product of the expectations of its parts [Martens and Grosse, 2020]. The Kronecker product has the property that its inverse is equal to the Kronecker product of its inverse parts, $(\boldsymbol{A} \otimes \boldsymbol{G})^{-1} = \boldsymbol{A}^{-1} \otimes \boldsymbol{G}^{-1}$. This is crucial for the LA, as the inverse GGN can be computed without storing the full matrix.

## 4 SPARSE-KFAC LAPLACE APPROXIMATIONS

In this paper, we propose a selection strategy for efficiently defining stochastic subsets in the KFAC LA, an illustration of which can be seen in Fig. 1. For the LA, we observed

that in small-scale MLPs, selecting the parameters with the highest $\ell_1$ norm performed better than random selection and selection based on variances as proposed in [Daxberger et al., 2022b] Appendix B. Only a small percentage of stochastic parameters was required to perform on par with full stochasticity if these were chosen based on their norm. Given the better performance of the largest $\ell_1$-norm selection strategy for MLPs we extend this concept into the space of Transformer models. The size of the transformer model applied to NLP problems, restricts the LA to using KFAC. The KFAC LA methodology Section 3 approximates the derivatives of a layer $i$ using the inputs $\hat{x}$ and output derivatives $\delta$ Eq. (16). Selecting a stochastic subset based on $\ell_1$-norm of the weights in the neural network is therefore not trivial as the GGN in a factorized form. Hence selecting the stochastic parameters requires consideration of the factorization procedure. The gradients of a selection of parameters from a weight matrix $\boldsymbol{W}$ can be factorized via the KFAC if they form a submatrix of $\boldsymbol{W}$, where the submatrix is formed by deleting rows or columns in $\boldsymbol{W}$. Freely selecting the parameters in $\boldsymbol{W}$ with the highest $\ell_1$-norm will most likely not form a submatrix.

In the S-KFAC method, we relax the requirement of only selecting the weights with the largest $\ell_1$ norms to a requirement where the largest $\ell_\infty$-norm for the rows $\boldsymbol{r}$ and columns $\boldsymbol{c}$ is selected. From the duality of the $\ell_\infty$ and $\ell_1$ norms Christensen [2010], we know that the largest $\ell_1$ norm parameters are placed in the rows and columns of $\boldsymbol{W}$ with the largest $\ell_\infty$ norm. We then construct the stochastic subset using the intersection between the rows and columns with the largest $\ell_\infty$-norm. As such a proxy for $\ell_1$-norm selection is done through the $\ell_\infty$-norm, where we can not guarantee all largest $\ell_1$-norm weights to be selected, but a considerable portion is included.

We write the linear layer formulation $l(\hat{x}) = \boldsymbol{W}\hat{x} = \boldsymbol{a}$ used in deriving the Fisher block approximation in Eqs. (15) and (16). We consider the sets of $\ell_\infty$ norms of the rows $\boldsymbol{r} = \{||\mathbf{W}_{i,:}||_\infty\}_{i=1}^n$ and columns $\boldsymbol{c} = \{||\mathbf{W}_{:,j}||_\infty\}_{j=1}^m$ in the weight matrix $\boldsymbol{W} \in \mathbb{R}^{n \times m}$. Importantly, since $n$ is the number of outputs and $m$ the number of inputs, $\boldsymbol{r}$ and $\boldsymbol{c}$ correspond to entries in $\boldsymbol{\delta}$ and $\hat{x}$ respectively.

A predefined percentage $p$ is defined as a hyperparameter for controlling the subset size, e.g. $p = 10$ would result in selecting 10% of the rows and columns with the largest $\ell_\infty$-norm values. We then select subsets of $\hat{x}$ and $\boldsymbol{\delta}$ by

$$\begin{aligned} \tilde{\boldsymbol{x}} &= \{\hat{\boldsymbol{x}}_j | c_j > P_p(\boldsymbol{c}), j \le m\}, \\ \tilde{\boldsymbol{\delta}} &= \{\boldsymbol{\delta}_i | r_i > P_p(\boldsymbol{r}), i \le n\}. \end{aligned} \quad (17)$$

Here $P_p : \boldsymbol{x} \to \mathbb{R}$ is the percentile operator for a percentile $p$. The Fisher block representing the selected stochastic subset is then constructed by

$$\mathcal{F}_i \approx \mathbb{E}[\tilde{\boldsymbol{x}}\tilde{\boldsymbol{x}}^T] \otimes \mathbb{E}[\tilde{\boldsymbol{\delta}}\tilde{\boldsymbol{\delta}}^T] = \tilde{\boldsymbol{A}} \otimes \tilde{\boldsymbol{G}}. \quad (18)$$

To relate this back to the specific weight matrix, the stochastic parameters become $\{\boldsymbol{W}_{i,j} | r_i > P_p(\boldsymbol{r}) \wedge c_j > P_p(\boldsymbol{c}), i \le n, c \le m\}$, i.e. the parameters lying in the intersection between the selected rows and columns. A visualization of a selected stochastic subset for a single transformer block is done in Fig. 1. The method is then extended to all layers, such that the stochastic subset is distributed over the entire network while controlling the subset size using the hyperparameter $p$.

In most cases, the linear mapping will be affine, i.e. including a bias term. For a bias $\boldsymbol{b} \in \mathbb{R}^n$ its derivative is given by $\nabla_{\boldsymbol{b}} \ln p(\boldsymbol{w}|\mathcal{D}) = \boldsymbol{\delta}$. As we are already storing $\tilde{\boldsymbol{\delta}}$, we make the corresponding elements in the bias stochastic too.

The selection made by the S-KFAC method ensures that the $\min(n, m) \times \frac{p}{100}$ largest $\ell_1$ norm parameters are included in the stochastic subset. This corresponds to selecting $\frac{pm}{100}$ input neurons and $\frac{pn}{100}$ output neurons and making the dense linear mapping between those stochastic.

Each input and output are chosen based on a single weight connecting them. Of course, with the possibility that multiple of the largest weights come from the same input or are going to the same output. Yet all weights between the chosen inputs and outputs are made stochastic. This causes the previously mentioned relaxation on choosing only the largest parameter weights, as we have no guarantee on the magnitude of the remaining weights.

We denote this strategy as Sparse-KFAC (S-KFAC) from the ensuing sparsity in the full covariance matrix Appendix A. Since each 'small' matrix in the KFAC approximation are $(n \times n)$ and $(m \times m)$, S-KFAC restricts the memory usage as the combined size of the matrices becomes $(\frac{np}{100})^2 + (\frac{mp}{100})^2 \ll n^2 + m^2$ for small percentages $p$. Thus, S-KFAC is lightweight in memory while capturing the important uncertainty from the parameter selection. Further, a large portion of the computational load in the KFAC LA method is contained in matrix-matrix multiplications for calculating the posterior covariance. The computational demand of matrix-matrix multiplications has cubic scaling and S-KFAC drastically lowers the matrix dimensionality. Thus given the same number of stochastic parameters in a partially stochastic model for the KFAC and the S-KFAC methods, S-KFAC will have a significantly lower computational demand at inference time.

## 5  EXPERIMENTS & METHODOLOGY

As the basis for this study is NLP tasks, we employ a pre-trained DistilBERT transformer encoder architecture[Sanh et al., 2020]. It is beyond the scope of the paper to provide a complete description of the Transformer architectures, and we refer to Vaswani et al. [2017] for the details. For this study, however, it is important to recall that the multi-head attention mech-

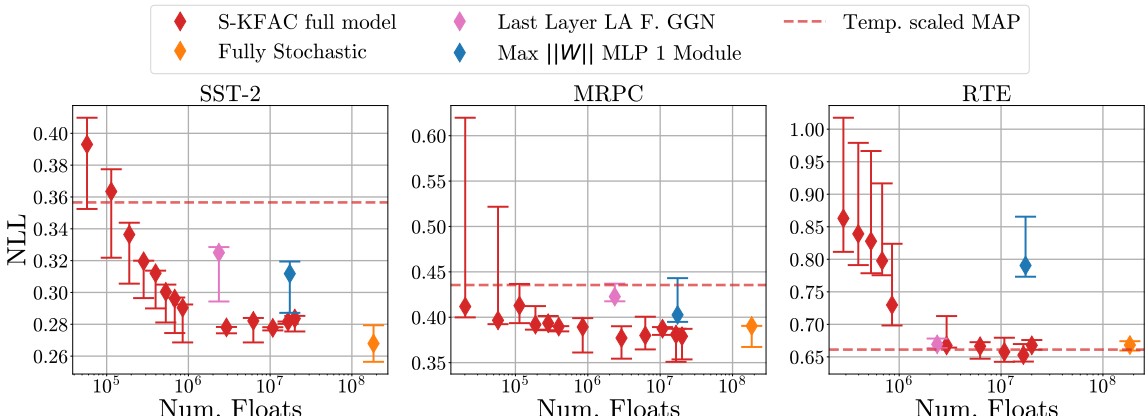

Figure 2: Ramping experiments using dense stochastic subparts of each weight matrix in the entire network over different percentages of stochasticity. We display the results over the three GLUE datasets SST-2, RTE, and MRPC. Points are median over 5 train/validation splits, and error bars are interquartile ranges. For each dataset, we compare with the median of the LLLA evaluated using the full GGN, the temperature-scaled MAP solution, and the MLP module with the largest $\|\boldsymbol{W}\|$.

anism [Vaswani et al., 2017] contains 4 affine mappings: $(\boldsymbol{W}_Q, \boldsymbol{b}_Q), (\boldsymbol{W}_K, \boldsymbol{b}_K), (\boldsymbol{W}_V, \boldsymbol{b}_V), (\boldsymbol{W}_O, \boldsymbol{b}_O) \in (\mathbb{R}^{n \times n}, \mathbb{R}^n)$, where for the distilBERT model $n = 768$. To complete the transformer block a one-layer feed-forward model or MLP is used containing affine mappings $(\boldsymbol{W}_{\text{in}}, \boldsymbol{b}_{\text{in}}) \in (\mathbb{R}^{n \times m}, \mathbb{R}^m)$ and $(\boldsymbol{W}_{\text{out}}, \boldsymbol{b}_{\text{out}}) \in (\mathbb{R}^{m \times n}, \mathbb{R}^n)$, where $m = 3072$. We use the terminology of 'modules' to refer to an affine mapping in either the attention or MLP block. This designation is motivated by the block structure of the KFAC approximation. We will therefore present results comparing performance between modules selected from either the MLP or attention blocks despite the varying number of parameters in them. A significant proportion of the learnable model parameters lie in the token embeddings (tokens are generated through the word-piece algorithm for DistilBERT [wor]). Which of these parameters are used is therefore dependent on the textual input and they are therefore never included in the stochastic subset. Additionally, when we calculate the percentage of stochastic parameters in the model, we do not count the token embeddings in the total number of parameters. We perform our experiments on three GLUE tasks: SST-2 [Socher et al., 2013], RTE [Dagan et al., 2006], and MRPC [Dolan and Brockett, 2005], all binary classification tasks. Information relating to the datasets is shown in Table 4. For each, we fine-tune 5 pre-trained DistilBERT models [dis] on 5 train/validation splits of the respective training sets, where we use the validation set for early stopping and hyperparameter tuning in for the respective methods for posterior inference. We use accuracy as the early stopping criterion. We keep the pre-allocated development sets as the test sets Wang et al. [2019]. Details can be found in Appendix C.1. Hence, all experiments are performed 5 times on all three datasets. We will refer to the resulting point estimates from the fine-tuning as the MAP solution $\boldsymbol{w}_{\text{MAP}}$. We mainly consider two hyperparameters of interest in the LA and SWAG

Table 1: The median accuracy, negative log-likelihood (NLL), and expected calibration error (ECE) for models using $\boldsymbol{w}_{\text{MAP}}$ as point estimates evaluated on the test sets.

| Metric | SST-2 | MRPC | RTE |
|---|---|---|---|
| NLL | 0.75 | 0.82 | 1.70 |
| Accuracy | 0.89 | 0.84 | 0.61 |
| ECE | 0.10 | 0.15 | 0.33 |

approximations. For the LA, it is the choice of prior variance, and for SWAG, it is the choice of learning rate. Both are estimated through cross-validation, with NLL on the validation set as the criterion. Additional details for SWAG and KFAC LA are shown in Appendix D.

## 5.1 INVESTIGATING THE HYPOTHESIS OF PARTIAL STOCHASTICITY IN NLP

Fig. 3 shows the results for the effects of partial stochasticity vs full stochasticity for the SWAG and KFAC LA posterior approximations on the MRPC and RTE datasets. For both approximations, we iteratively increase the percentage of stochastic parameters in the model, where the stochastic parameters are chosen uniformly at random. We observe that partial stochasticity performs better than the MAP solution for even small percentages of stochastic parameters. Furthermore, we see that partial stochasticity tends to either outperform or perform equally to its fully stochastic variant. The results for the SST-2 dataset can be seen in Appendix E.

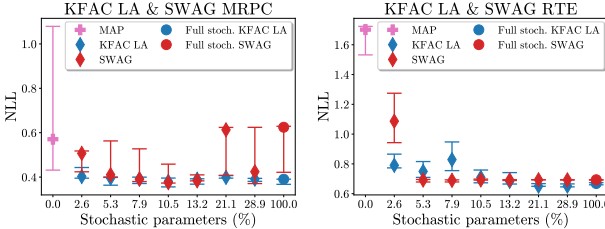

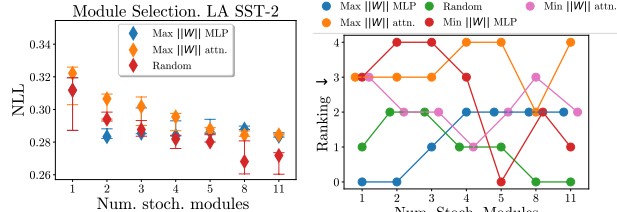

Figure 3: Experiment varying the percentages of stochastic parameters in the KFAC LA and SWAG approximations for posterior inference on the MRPC and RTE datasets.

Figure 4: **Left**: Max $\|\boldsymbol{W}\|$ ramping experiments for the attention and feed-forward modules, along with random selection on the SST-2 dataset, using LA as the posterior approximation. **Right**: Rankings of each modular stochastic subset selection scheme across the 3 datasets/tasks for each number of stochastic modules, where 0 is optimal.

## 5.2 STOCHASTIC MODULE SELECTION

To compare the stochastic subset selection frameworks, we consider 5 ramping schemes. Of these, 3 are modular, i.e. entire modules or linear mappings, including bias, on which to induce stochasticity are selected. Experiments conducted with small-scale feed-forward models indicated that a norm-based selection strategy on individual parameters improved results Appendix B. We extend this to a modular basis by using the operator norm. More specifically, we set $\|\boldsymbol{W}\| = \sigma_1(\boldsymbol{W})$, where $\boldsymbol{W}$ is a linear operator, e.g. one of the weight matrices in the model, and $\sigma_1$ is its largest singular value [Christensen, 2010].

In the DistilBERT models, we observed that the operator norms of the weight matrices in the MLP modules were significantly larger than the ones in the self-attention mechanism Appendix C and Fig. 11. Thus, we consider three main module ramping experiments: 1) Iteratively adding the next module with the largest operator norm from the MLP mappings to the stochastic subset, 2) the same procedure as in 1) but from the attention mechanism, 3) Random selection over the entire model. Note here that we include the classifier in the MLP grouping. In Fig. 4, we present the results from these modular ramping schemes on the SST-2 dataset and using the KFAC LA.

The MAP results can be seen in Table 1. We observe that the most immediate drop in NLL happens for the max $\|\boldsymbol{W}\|$ MLP module strategy. However, note that those linear layers contain four times as many parameters as a linear layer in the attention mechanism Section 5. Additionally, random selection significantly outperforms norm-based selection from the inclusion of 8 layers. Similar tendencies were observed for the other two tasks, see Appendix E.

To decorrelate the performance increase for the max operator norm selection strategy from the additional stochastic parameters, we perform similar experiments by ramping from the minimum rather than the maximum Appendix E. As the focus is to ascertain optimal selection strategies, we display the rankings of these 5 operator norm-related experiments - including random - in Fig. 4. For each dataset, we compute the ranking of the 5 selection schemes. For each

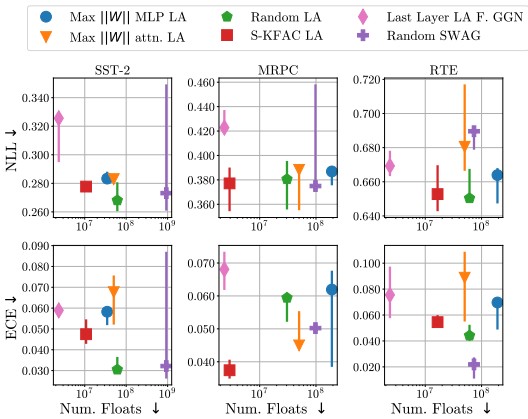

Figure 5: Comparison of negative log-likelihood (NLL) and expected calibration error (ECE) as a function of memory footprint for each ramping scheme for each dataset. For each ramping scheme, we select the best-performing level of stochasticity in terms of NLL. Error bars are interquartile ranges across 5 train/validation splits. The best-performing SWAG ramping scheme is included for comparison.

module, we present the median ranking of the respective method. Note that to have a 0 ranking, the method must have the highest performance on at least 2 of the datasets. We observe that selecting MLP modules with the largest operator norm is the highest-performing strategy across the 3 tasks for 1, 2, and 3 modules. Comparatively, selecting the minimum is the worst-performing strategy in that range.

Finally, we observe that random selection, after the inclusion of 5 layers, outperforms the other methods. The observation that random selection outperforms the other selection heuristics for a sufficiently large number of stochastic modules is not necessarily intuitive. It could indicate that each mechanism (MLP & attention) contributes differently to the uncertainty of the model. And that it is beneficial to capture both 'types' rather than thinking only in terms of the number of stochastic parameters.

Table 2: Results for ramping experiments conducted on the three GLUE tasks. For these experiments, we restrict the number of stochastic parameters in the model to be approx. 10 pct. of all model parameters. The NLL and ECE are reported on all tasks for the MAP solution, temperature scaling, fully stochastic KFAC LA, and SWAG. We show $\pm$ standard errors, where $\pm 0.00$ indicates the uncertainty is below 0.005. The LLLA is calculated using the full GGN approximation. The best overall performance for each dataset is highlighted in **bold**, and the best partially stochastic method is highlighted with underline.

| Methods | SST-2 | | MRPC | | RTE | |
|---|---|---|---|---|---|---|
| | NLL↓ | ECE↓ | NLL↓ | ECE↓ | NLL↓ | ECE↓ |
| MAP | $0.74 \pm 0.10$ | $0.11 \pm 0.01$ | $0.57 \pm 0.34$ | $0.14 \pm 0.04$ | $1.71 \pm 0.23$ | $0.33 \pm 0.05$ |
| **LA** | | | | | | |
| Temp. Scaled | $0.36 \pm 0.02$ | $0.06 \pm 0.00$ | $0.44 \pm 0.01$ | $0.05 \pm 0.02$ | $0.66 \pm 0.01$ | $0.13 \pm 0.04$ |
| Min $\|\boldsymbol{W}\|$ MLP | $0.33 \pm 0.02$ | $0.06 \pm 0.01$ | $0.42 \pm 0.01$ | $0.07 \pm 0.01$ | $0.67 \pm 0.01$ | $0.08 \pm 0.03$ |
| Max $\|\boldsymbol{W}\|$ attn. | $0.29 \pm 0.00$ | $0.04 \pm 0.00$ | $0.39 \pm 0.02$ | $\underline{\mathbf{0.04}} \pm 0.01$ | $0.81 \pm 0.11$ | $0.13 \pm 0.03$ |
| Min $\|\boldsymbol{W}\|$ attn. | $\underline{0.28} \pm 0.00$ | $0.04 \pm 0.01$ | $0.39 \pm 0.02$ | $0.05 \pm 0.00$ | $0.72 \pm 0.05$ | $0.11 \pm 0.02$ |
| Random | $\underline{0.28} \pm 0.01$ | $0.04 \pm 0.01$ | $0.38 \pm 0.02$ | $0.05 \pm 0.01$ | $0.71 \pm 0.05$ | $0.14 \pm 0.03$ |
| S-KFAC | $\underline{0.28} \pm 0.00$ | $0.04 \pm 0.01$ | $\underline{\mathbf{0.37}} \pm 0.02$ | $\underline{\mathbf{0.04}} \pm 0.00$ | $\underline{\mathbf{0.65}} \pm 0.01$ | $0.05 \pm 0.01$ |
| Last Layer | $0.33 \pm 0.02$ | $0.06 \pm 0.00$ | $0.42 \pm 0.01$ | $0.07 \pm 0.01$ | $0.67 \pm 0.01$ | $0.08 \pm 0.03$ |
| **SWAG** | | | | | | |
| Max $\|\boldsymbol{W}\|$ MLP | $0.52 \pm 0.06$ | $0.09 \pm 0.01$ | $0.51 \pm 0.12$ | $0.09 \pm 0.03$ | $1.09 \pm 0.20$ | $0.28 \pm 0.03$ |
| Max $\|\boldsymbol{W}\|$ attn. | $0.39 \pm 0.08$ | $\underline{\mathbf{0.03}} \pm 0.04$ | $0.42 \pm 0.18$ | $0.07 \pm 0.05$ | $0.74 \pm 0.02$ | $0.11 \pm 0.03$ |
| Random | $0.32 \pm 0.10$ | $0.04 \pm 0.02$ | $0.39 \pm 0.06$ | $0.05 \pm 0.01$ | $0.69 \pm 0.01$ | $\underline{\mathbf{0.02}} \pm 0.02$ |
| Sublayer $\ell_1$ | $0.39 \pm 0.13$ | $0.07 \pm 0.03$ | $0.44 \pm 0.04$ | $\underline{\mathbf{0.04}} \pm 0.02$ | $0.69 \pm 0.00$ | $\underline{\mathbf{0.02}} \pm 0.01$ |
| Fully Stoch. (**LA**) | $\mathbf{0.27} \pm 0.01$ | $\mathbf{0.03} \pm 0.00$ | $0.39 \pm 0.01$ | $0.06 \pm 0.00$ | $0.66 \pm 0.01$ | $0.06 \pm 0.01$ |
| Fully Stoch. (**SWAG**) | $\mathbf{0.27} \pm 0.06$ | $\mathbf{0.03} \pm 0.03$ | $0.62 \pm 0.11$ | $0.07 \pm 0.02$ | $0.69 \pm 0.00$ | $0.05 \pm 0.02$ |

## 5.3 SPARSE-KFAC LAPLACE

We now assess the performance of the S-KFAC selection scheme. We conduct three ramping experiments, where we vary the percentage of stochastic parameters in the model according to the methodology described in Section 4. The results can be seen in Fig. 2. We compare the results with the commonly used Last Layer LA (LLLA) [Daxberger et al., 2022a] with full GGN approximation, Temperature scaling [Guo et al., 2017], max $\|\boldsymbol{W}\|$ MLP with one module and full stochasticity. Note that the number of floats corresponds to percentages as seen in Appendix A and Table 3.

On the SST-2 and MRPC, there exist several percentages of stochasticity for which S-KFAC requires less memory than the LLLA yet yields a significantly lower NLL. Additionally, it repeatedly outperforms the single module with max $\|\boldsymbol{W}\|$, again while requiring less memory. The exception is the RTE dataset, in which temperature scaling and the LLLA yield comparable performance to S-KFAC. The single MLP module outperforms LLLA on 2 out of 3 datasets. However, even a single module requires more memory than S-KFAC, with approx. 10 pct of the model parameters.

Moreover, the percentage of stochastic parameters (Num. Floats) at which the best NLL is achieved for the S-KFAC method varies across the three datasets. Hence, applications of the methodology will require a sweep through cross-

validation to estimate the percentage for optimal performance. Recall that given percentages $p_1 < p_2$ the following will hold for the stochastic subsets in the S-KFAC method $\boldsymbol{w}_{S,p_1} \subset \boldsymbol{w}_{S,p_2}$. Therefore, the amount of uncertainty captured by the S-KFAC using $p_2$ is unlikely to be significantly lower than by using $p_1$. We hypothesize that this near-convex behavior may assist the estimation of p.

## 5.4 SELECTION STRATEGIES FOR SWAG & LA

Fig. 5 compares the S-KFAC, the three LA modular selection schemes, and random module selection for the SWAG approximation. For each selection strategy, we select the level of stochasticity yielding the lowest validation NLL. The random selection scheme was found to perform best for SWAG Table 2. We observe that the best performance is often found using random selection on a modular level for the Laplace approximation. However, it is rarely significantly better than the S-KFAC selection strategy, demanding approximately 7 times more memory on average. Additionally, we observe that SWAG displays similar performance to LA but with a significantly higher computational cost.

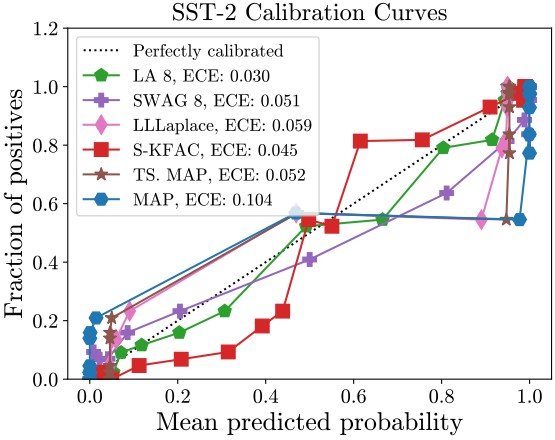

Figure 6: Calibration curves on SST-2. The median expected calibration error (ECE) across 5 train/validation splits is displayed for each method. 20 bins are placed based on the prediction probability distribution rather than uniformly. The LA and SWAG are shown for 8 randomly selected stochastic linear layers. S-KFAC is shown for $1.588\%$ of parameters being stochastic.

In Table 2, we present results for all subset selection strategies. However, we limit the percentage of stochastic parameters in the model to approximately 10 pct. of the full parameter set. We additionally compare with full stochasticity for both LA and SWAG. In Fig. 5, we observed that SWAG was quite competitive with the LA when no restrictions were placed on the number of stochastic parameters. However, when limited to approximately 10 pct. It yields significantly higher NLL than all LA selection schemes. Furthermore, we also observe that S-KFAC shows the largest improvement over the MAP solution of all methods in terms of NLL. While SWAG often attains the lowest ECE, it does not manage to retain performance on NLL. Finally, on 2 out of three tasks, full model stochasticity yields worse performance than the partial schemes for SWAG and LA.

Fig. 6 shows the calibration curves for a selection of the subset selection methods. The curves for the other datasets can be seen in Appendix G. The three high-performing methods, random selection LA and SWAG, along with S-KFAC, are all observed to alter the underlying structure of the predictive distribution. The LLLA and temperature scaling appear to shift inwards the extreme probabilities, approximately down to the model accuracy Table 1. Squeezing the probabilities inwards lowers the ECE. However, it does facilitate improvement in post hoc applications such as thresholding.

# 6 CONCLUSION

In this paper, we perform a detailed study of the effects of Bayesian inference for a partially stochastic subset of parameters in transformer models for NLP tasks. We validate the efficacy of subnetwork inference presented in Daxberger et al. [2022b], Sharma et al. [2023] for transformers across three GLUE tasks. The Subnetwork inference methodology is evaluated on the posterior approximation methods Stochastic weight averaging - Gaussian (SWAG) and the Laplace approximation (LA). We find that stochastic subset inference unequivocally outperforms the MAP solutions and generally displays similar or improved performance compared to fully stochastic variants.

We propose and evaluate heuristics for selecting the size of the stochastic subset on a modular level. We selected linear mappings to include in the stochastic subset from the MLP and attention blocks based on their operator norms. We found that norm-based selection yielded the best performance for small stochastic subsets. For larger numbers of modules, a random selection scheme is dominant. Indicating that the MLP and attention components contribute differently to the uncertainty of the model. Given this finding, we conclude that a homogeneous distribution of stochastic parameters is preferred.

We proposed a novel method, Sparse-KFAC, for selecting stochastic subsets by creating dense stochastic substructures in all linear mappings in the model. We found that Sparse-KFAC invariantly yielded competitive or higher performance than all other selection strategies while requiring orders of magnitude fewer parameters. Additionally, we found that when the stochastic subset was limited to 10 pct. of the parameters in the model, Sparse-KFAC outperformed all other methods, including full stochasticity, on two out of three tasks. However, the method introduces an additional hyperparameter defining the percentage of stochastic parameters in each affine mapping. We observed different 'speeds of convergence' towards the optimal across the three datasets. Hence employment of the method requires estimating the parameter through cross-validation. However, the fact that Sparse-KFAC is fully defined through the choice of a percentage of stochastic weights also greatly simplifies the selection process. For example, given memory limitations that allow for a certain percentage $P$ of stochastic parameters. One can achieve this through approximately $n$ MLP modules, $4n$ attention modules, or a combination. The performance of each of these module combinations can vary greatly without clear prior indicators. Hence, the selection process requires an exhaustive search. In Sparse-KFAC $P$ pct. of the parameters can simply be selected optionally testing if lower percentages yield similar performance. Sparse-KFAC partially decouples the size of the stochastic subset from the width of the model. This is highly relevant as modern transformer architectures are increasingly wide. As such, the Sparse-KFAC method is readily extendable to larger models, for which stochastic subset selection is an interesting avenue for further research.

---

Source code: https://github.com/GustavAls/PartialNLP

**Acknowledgements**

This work is supported by Danish Pioneer Centre for AI (aicentre.dk), DNRF grant number P1.

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

# Towards Scalable Bayesian Transformers: Investigating stochastic subset selection for NLP
## (Supplementary Material)

Peter J.T. Kampen[1,*]          Gustav R.S. Als[1,*]          Michael Riis Andersen[1]

[1]DTU Compute, Technical University of Denmark, Kongens Lyngby, Denmark
[*]These authors contributed equally,

## A    SPARSE-KFAC

KFAC approximation derivation from Eq. (16)

$$
\begin{aligned}
\mathcal{F}_i &= \mathbb{E}_{p(\mathcal{D})} \left[ \nabla_{\boldsymbol{w}^{(i)}} \ln p(\mathcal{D}|\boldsymbol{w}) \nabla_{\boldsymbol{w}^{(i)}} \ln p(\mathcal{D}|\boldsymbol{w})^T \right] \\
&= \mathbb{E}_{p(\mathcal{D})} \left[ (\hat{\boldsymbol{x}} \otimes \boldsymbol{\delta})(\hat{\boldsymbol{x}} \otimes \boldsymbol{\delta})^T \right] \\
&= \mathbb{E}_{p(\mathcal{D})} \left[ (\hat{\boldsymbol{x}} \otimes \boldsymbol{\delta})(\hat{\boldsymbol{x}}^T \otimes \boldsymbol{\delta}^T) \right] \\
&= \mathbb{E}_{p(\mathcal{D})} \left[ \hat{\boldsymbol{x}}\hat{\boldsymbol{x}}^T \otimes \boldsymbol{\delta}\boldsymbol{\delta}^T \right] \\
&\approx \mathbb{E}[\hat{\boldsymbol{x}}\hat{\boldsymbol{x}}^T] \otimes \mathbb{E}[\boldsymbol{\delta}\boldsymbol{\delta}^T] \\
&= \boldsymbol{A} \otimes \boldsymbol{G}.
\end{aligned}
$$

### A.1    COVARIANCE STRUCTURE OF SPARSE-KFAC

The Sparse-KFAC methodology introduced in this paper aims to select substructures of the KFAC Hessian approximation to sparsely distribute stochastic parameters across the neural network. The selection is done by choosing specific inputs and doing a fully connected mapping to specific outputs. To visualize the covariance structure of this mechanism, a series of stochastic percentiles are chosen and used to generate an S-KFAC Hessian approximation. The stochastic weights are shown in white, and the deterministic parameters are shown in black Fig. 7. The fraction of stochastic parameters compared with the total number of parameters in the selected linear layers (modules) can be described by $g_P = p^2/100^2, 0 < p \leq 100$. $g_P = 0.5$ indicates that half of the total number of parameters are selected for the stochastic subset.

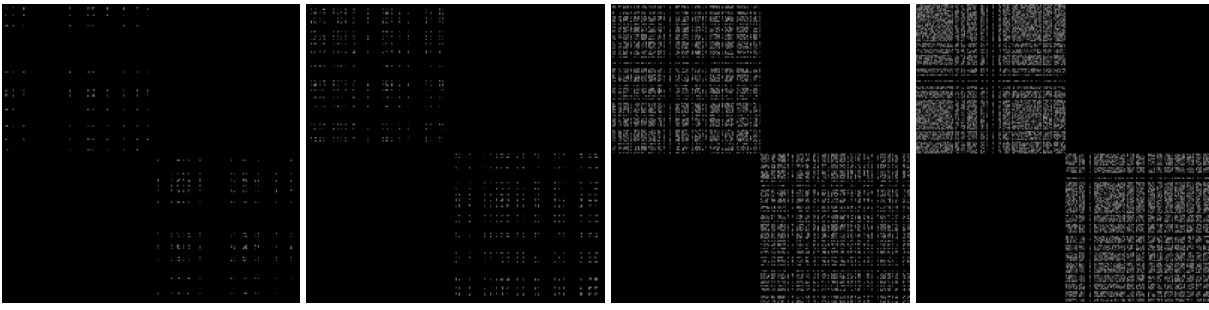

Figure 7: Sparse covariance structures visualized for varying percentiles, and selection according to the Sparse-KFAC methodology described in Eqs. (17) and (18). A two-layer neural network with a 100-layer width is shown. The selected percentiles from left to right are $p = 40$, $p = 50$, $p = 80$, and $p = 90$.

## A.2 PERCENTILE RAMPING

Table 3: The percentile ramping schemes $\boldsymbol{w}_{S,p}$ used in this study for the Sublayer experiments. This includes S-KFAC on the full model, S-KFAC on the partial modules, and the SWAG $\ell_1$ ramping scheme. Percentages for S-KFAC are then used as explained in Section 3.2 and correspond to the percentage of rows and columns in the weight matrices, not the percentage of parameters.

| Ramping Scheme | $\boldsymbol{w}_{S,p}$ |
|---|---|
| SWAG | $p \in \{0, 1, 12, 24, 36, 48, 60, 100\}$ |
| S-KFAC | $p \in \{0, 1.0, 1.8, 2.5, 3.2, 3.9, 4.6, 5.4, 6.1, 6.8, 12.6, 18.4, 24.2, 30, 33, 100\}$ |

## A.3 SPECIFIC MODULE RAMPING

Here are the results where we specify the modules on which to apply the Sparse-KFAC selection. For each dataset/task, we specify a selection of modules, e.g. the 2 MLP modules with the largest operator norm. We then apply the S-KFAc selection strategy to only those modules, leaving the remaining modules deterministic. This is done to show that S-KFAC can be applied post hoc even if one has first found a selection of modules that one wants to induce stochasticity on. The results show that S-KFAC captures the uncertainty in a model around the inclusion of 25 pct of the stochastic parameters in all three cases.

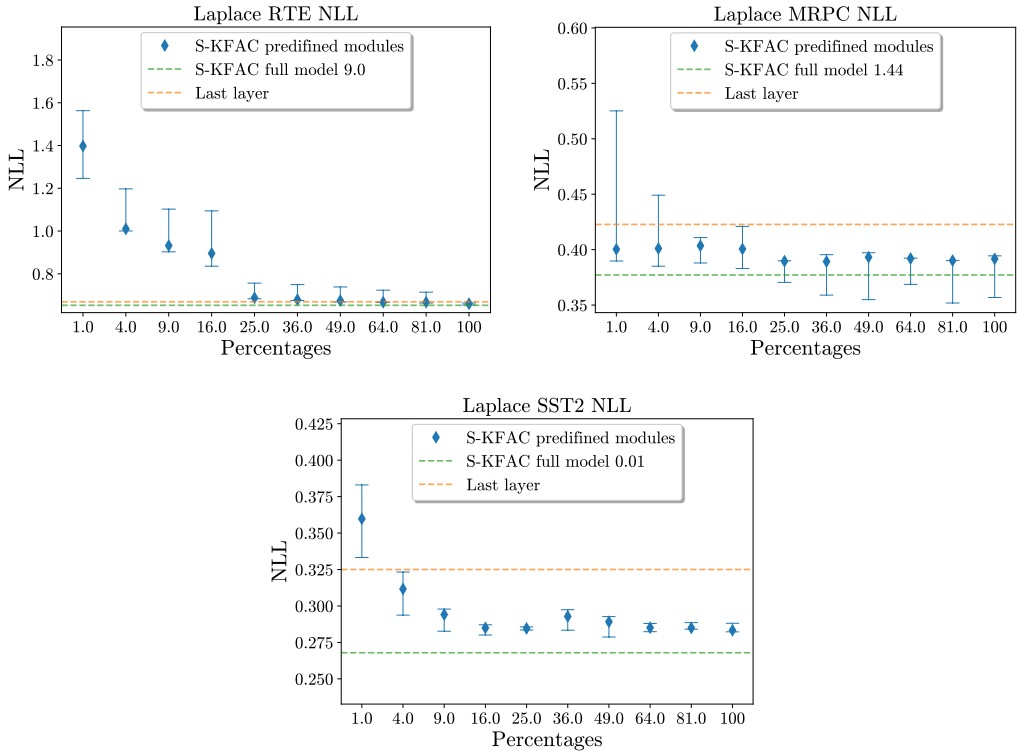

Figure 8: Results for iteratively increasing stochasticity through S-KFAC on specified modules across the three tasks. The 100 pct are full model stochasticity and conform to points seen for 2 or three modules in the Appendix E. The results show that S-KFAC captures the uncertainty in a model around the inclusion of 25 pct of the stochastic parameters in all three cases.

### A.4 RAMPING A SUBSET OF LINEAR MAPPINGS

# B UCI RESULTS

SWAG and LA results are presented on the three UCI regression datasets: Boston, Energy, and Yacht.

## B.1 LAPLACE AND SWAG ON UCI

To verify the efficacy of partially stochastic Bayesian neural networks, experimentation is done for small-scale regression problems. SWAG and the LA are tested across three UCI regression datasets. In Fig. 9, the median NLL is displayed for all three datasets where the pct. of stochastic parameters is varied. The MAP and fully stochastic performance are included for comparison. The Yacht dataset is shown with and without the MAP solution, so variation between subsets can be interpreted.

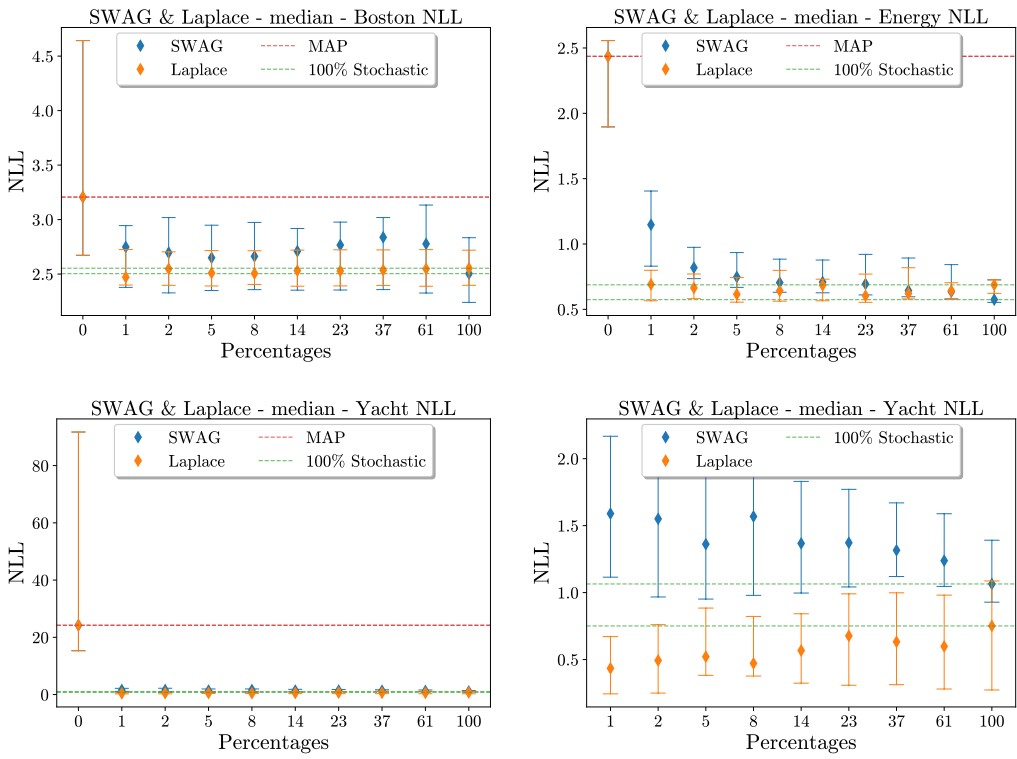

Figure 9: The median Negative log-likelihood (NLL) and its interquartile range of the posterior approximation strategies SWAG and Laplace across 15 train-test splits, compared on the UCI datasets Boston, Energy, and Yacht. Each method is fitted with a percentage of stochastic parameters shown on the x-axis and evaluated on the test set. The MAP solution and 100% stochastic solution are both shown with a red and green line, respectively. The Yacht results are shown with and without the MAP solution such that the inter-percentile NLL variation can be seen clearly.

## B.2 NORM VS VARIANCE BASED SELECTION

In Daxberger et al. [2022b], it is argued that a parameter selection strategy based on choosing the parameters with the largest marginal variances is proposed, arguing that this is favorable for closely approximating the posterior predictive distribution. In Sharma et al. [2023], the $\ell_1$-norm is instead used as a proxy for selecting the optimal stochastic subset. We compare the two methodologies for the three UCI regression datasets, Boston, Energy, and Yacht, to ascertain the best-performing subset selection strategy Fig. 10.

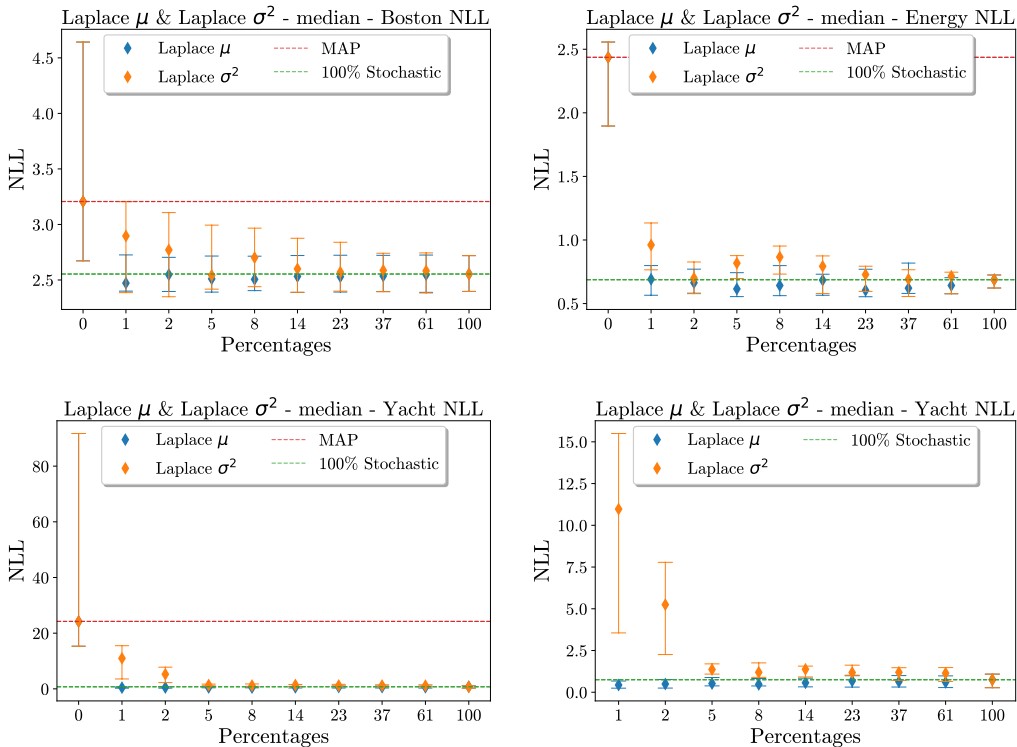

Figure 10: A comparison between two stochastic subset selection heuristics. Laplace $\mu$ chooses the highest parameter magnitudes based on $\boldsymbol{w}_{MAP}$. Laplace $\sigma^2$ follows Daxberger et al. Daxberger et al. [2022b] and selects a stochastic subset selection based on largest marginal variances, diag $\mathcal{G}^{-1}$. The median NLL and its interquartile range are shown over 15 runs for the UCI datasets Boston, Energy, and Yacht. Yacht is shown with and without MAP. The percentage of parameters modeled as stochastic is shown on the x-axis.

Fig. 10 indicates that norm-based subset selection strategy is superior for low percentiles. For Boston and Energy, the median of the two methods is not significantly different. However, on the Yacht dataset, the percentiles $p \in \{1, 2, 5, 8\}$ all show significant improvement by using the $\ell_1$-norm as compared with the marginal variance approach.

## C   EXPERIMENTS BACKGROUND

We extend the parameter magnitude based stochastic subset selection of Sharma et al. [2023] to transformers models by computing the operator norm over the networks modules, characterized by their singular values. To gain an understanding of the magnitude of the singular values over the entire DistilBERT architecture [Sanh et al., 2020] a boxplot is done of the singular values for the self-attention and MLP modules Fig. 11.

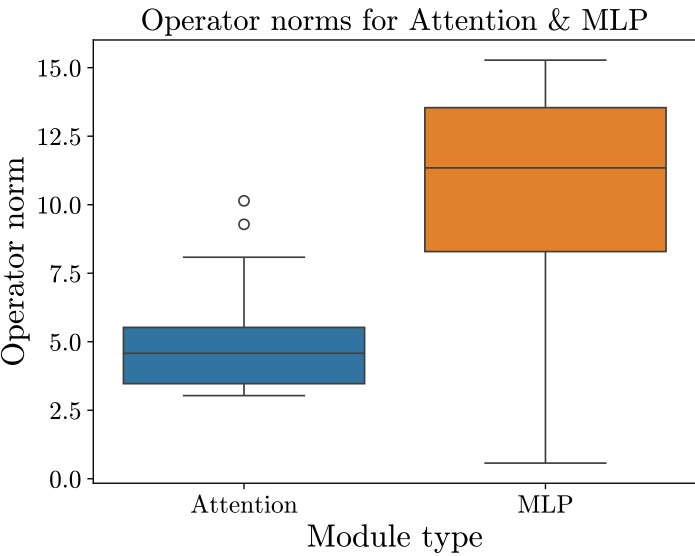

Figure 11: Boxplots showing the distribution of the operator norms of the weight matrices in the two different types of modules, namely self-attention, denoted attention, and MLP modules. We use the operator norm defined by the largest singular value. Bias terms are not included.

Interestingly, from Fig. 11, the singular values of the MLP modules are distributed substantially higher than those of the self-attention modules. Therefore, if the modules with the largest operator norm are selected indiscriminately the MLP modules will be included in the stochastic subset first. Additionally, the variance of the MLP singular values is greater than for self-attention. This

### C.1 FINE-TUNING DISTILBERT

In Table 4, information on the GLUE datasets investigated in this paper is presented.

Table 4: Information on the Natural Language Processing datasets selected for this study. The datasets and tasks selected are Sentiment Classification (SST-2 [Socher et al., 2013] Paraphrase Identification (MRPC [Dolan and Brockett, 2005]), and Natural Language Inference (RTE [Dagan et al., 2006]).

| Dataset | SST-2 | MRPC | RTE |
|---|---|---|---|
| Train size | $60,600$ | $3,300$ | 2240 |
| Val size | 6700 | 370 | 250 |
| Test (dev) size | 872 | 408 | 277 |
| Train/val class dist. ($true/false$) | $55.8\%/44.2\%$ | $67.4\%/32.6\%$ | $50.2\%/49.8\%$ |
| Test/dev class dist. ($true/false$) | $55.8\%/44.2\%$ | $68.4\%/31.6\%$ | $52.7\%/47.3\%$ |
| Max Sequence length | 268 | 226 | 1400 |

In Table 5, we present the hyperparameters chosen for fine-tuning the distilBERT model.

Table 5: The Hyperparameter configuration used for training DistilBERT (`distil-bert-uncased`) from the `transformers` *Huggingface* library.

| Hyperparameter | Description |
| --- | --- |
| Number of epochs | 10 |
| Batch size | 16 |
| Optimizer | AdamW |
| learning-rate | $5 \cdot 10^{-5}$ |
| $\beta_1$ | 0.9 |
| $\beta_2$ | 0.999 |
| FF dropout | 0.1 |
| Attention dropout | 0.1 |
| Sequence classifier dropout | 0.2 |

## D   IMPLEMENTATION DETAILS: LAPLACE APPROXIMATION AND SWAG

The configurations used when fitting the KFAC LA and SWAG are listed in Table 6.

Table 6: Training configurations for the Laplace and SWAG approximations. Common hyperparameters are noted in the top, SWAG-specific ones in the middle, and finally, Laplace

| Hyperparameter | Description or Value |
| --- | --- |
| Likelihood | Categorical (Cross-Entropy) |
| $\boldsymbol{w}_S$ Num. Modules | $[0, 1, 2, 3, 4, 5, 8, 11, 17, 38]$ |
| Batch Size | 16 |
| **SWAG** | |
| Learning rate sweep | $\left[10^{-3}, 10^{-2}, 5 \cdot 10^{-2}, 10^{-1}\right]$ |
| Optimizer | SGD |
| Momentum | 0.9 |
| Num. optim. steps sweep | 400 |
| Num. optim. steps final | 2000 |
| Num. Columns in D | 20 |
| Iterations between snapshots | 5 |
| Num. MC Samples | 50 |
| Sublayer $\boldsymbol{w}_{S,p}$ | $p \in [1, 12, 24, 36, 48, 60, 100]$ |
| **Laplace** | |
| Prior precision sweep | Equidistant in logspace in $[10^{-1}, 10^3]$ |

A validation set is used for tuning hyperparameters in both methods. For the LA the prior precision is tuned, and for SWAG, the learning rate.

## E   RAMPING EXPERIMENTS

In Fig. 12, we present the figure on partial stochasticity vs full and MAP for the SST-2 dataset, corresponding to the results shown in Fig. 4 for the MRPC and RTE tasks.

In Section 5.2, ramping experiments are shown for random selection and using the maximum operator norm on MLP and attention modules as heuristics for selecting modules for a stochastic subset. In Fig. 13, the same ramping experiments are shown for the RTE and MRPC datasets. **Left**: the MAP performance is included, **Right** it is excluded such that variation between subset size performance can be interpreted. The performance of a full GGN LLLA is highlighted by a green dotted line.

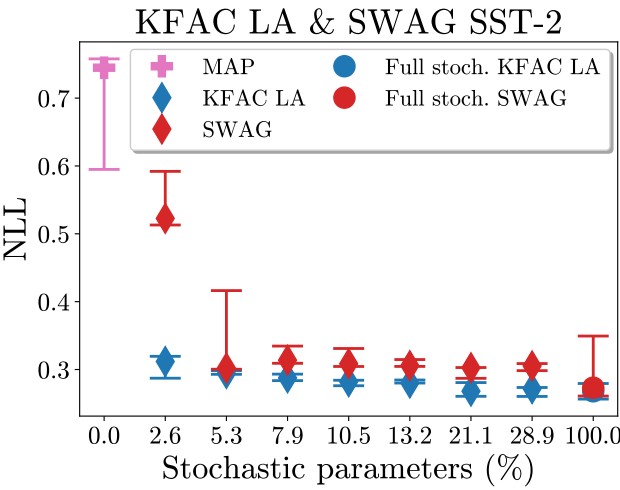

Figure 12: Results for the SST-2 dataset corresponding to the results presented in Fig. 4

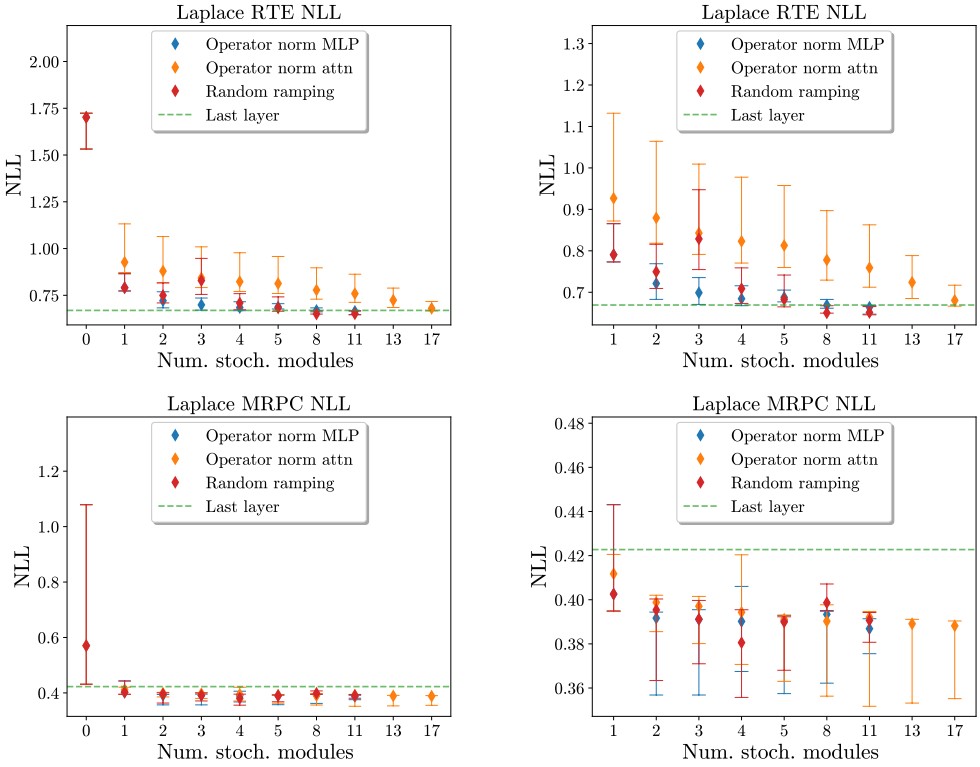

Figure 13: Median NLL of the Laplace approximation as a function of the number of modules selected under the three ramping schemes; random module selection, max operator norm/max operator norm MLP selection, and max operator norm of the attention modules. The median is taken over 5 train/val splits and evaluated on the RTE (top) and MRPC (bottom) test sets. The lines/ranges are interquartile. The left figure includes the MAP evaluation and the right without it. A line showing the performance of 'last layer Laplace' with full GGN approximation is included.

In Fig. 14, ramping experiments are shown for the minimum and maximum operator norm $\|\boldsymbol{w}\|$ are shown for the three GLUE datasets SST-2, RTE, and MRPC.

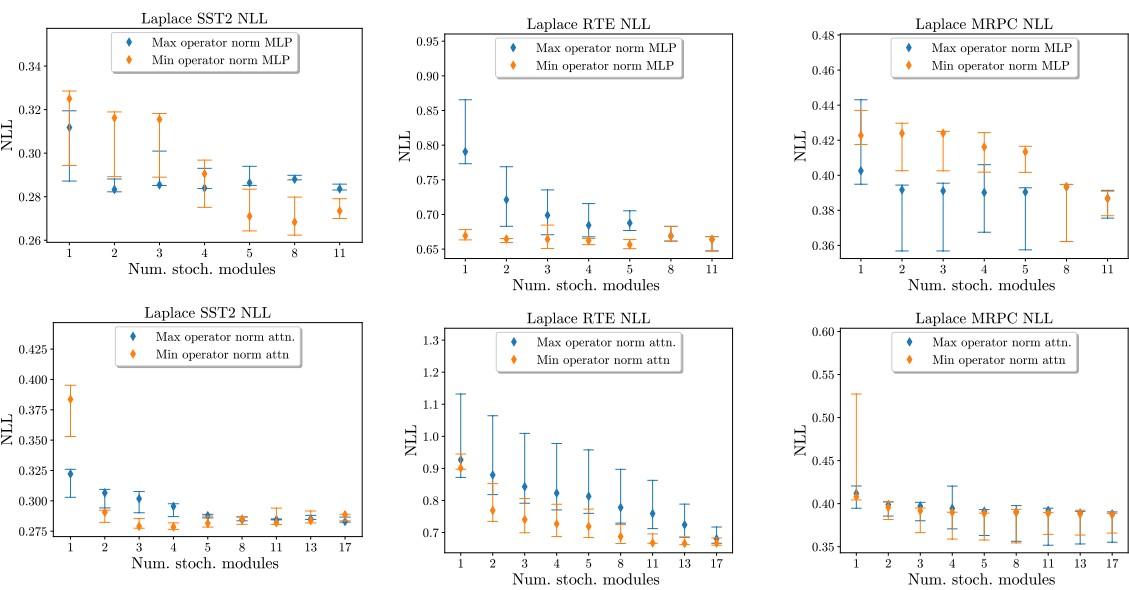

Figure 14: Median NLL results for the Laplace approximation for maximum and minimum operator norm ramping for MLP modules - left - and attention modules - right - across 5 train/val splits and evaluated on the test set for the SST-2 (top) RTE (middle) and MRPC (bottom) datasets. The shown ranges are interquartile.

# F SWAG RAMPING EXPERIMENTS

In this section, we briefly present the results seen for SWAG with the corresponding ramping experiments as conducted for the Laplace approximation. Namely: Max operator norm MLP, Max operator norm attention, and random ramping.

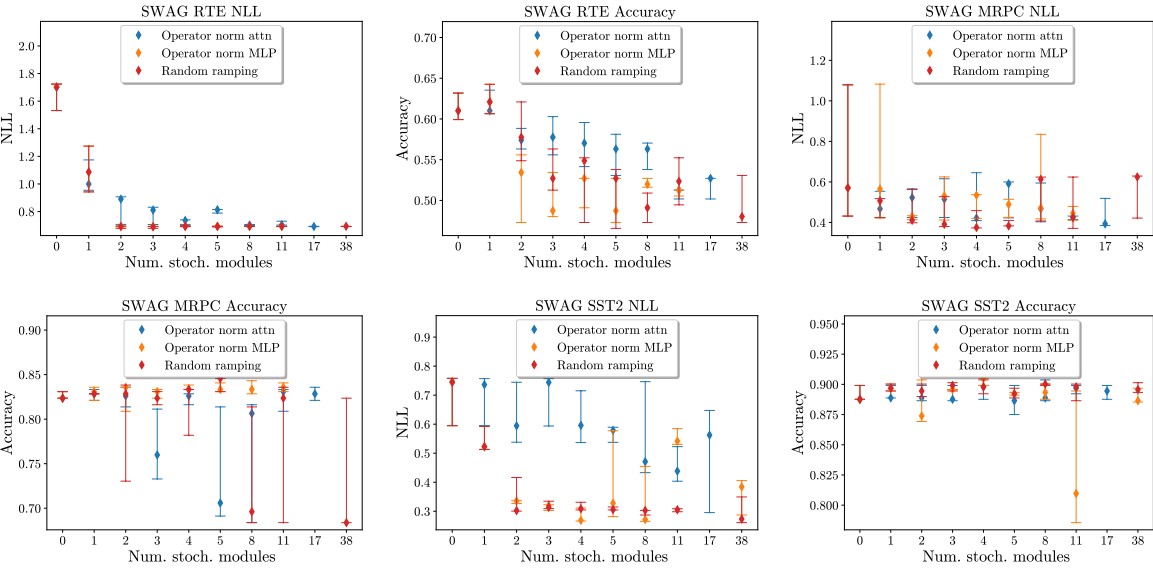

Figure 15: SWAG ramping experiments across the three GLUE tasks for the ramping schemes: Max operator norm MLP, Max operator norm attention, and random ramping. All models are optimized to minimize the NLL, while disregarding the accuracy. Hence the perhaps odd drops in performance at certain ranges of stochastic modules.

## F.1 SUBLAYER RAMPING IN SWAG

In Fig. 15 we present the results for ramping stochastic subsets on a modular level for the SWAG method. Here, an investigation into parameter-specific ramping is done. Using the $\ell_1$-norm as a heuristic, the ramping scheme is conducted

such $p$ pct. of the parameters with the largest norm are added to the stochastic subset. For comparison, the subsets with the lowest NLL are shown for the random and operator norm ramping schemes. Operator norm is shown for both MLP and attention module ramping.

The last SWAG results for the $\ell^1$ norm ramping scheme can be seen in Fig. 16.

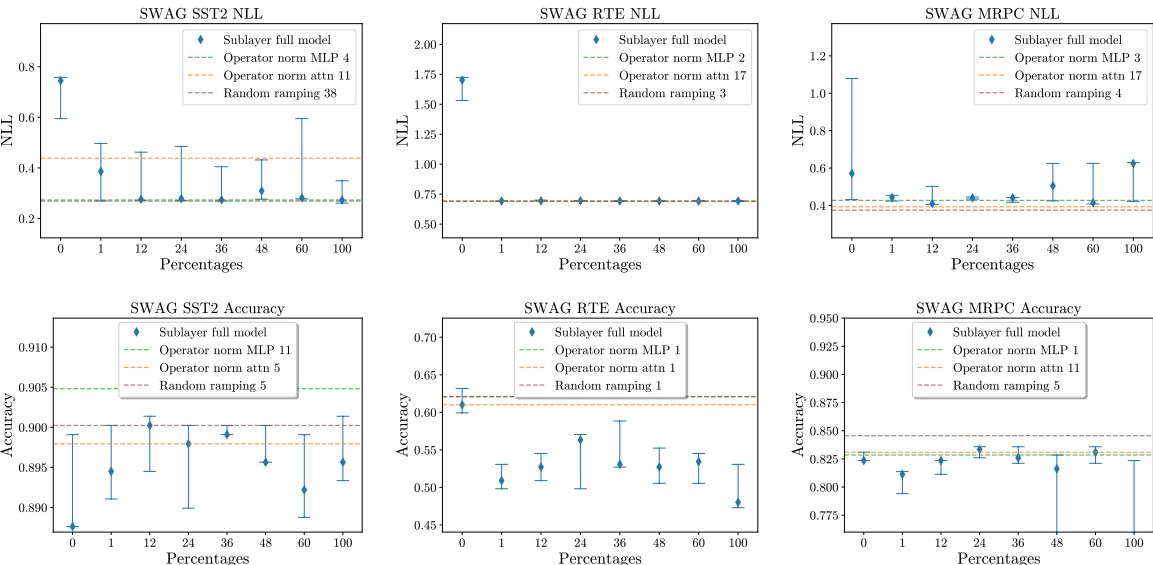

Figure 16: Median NLL and accuracy as a function of the model's number/percentage of stochastic parameters, displayed for SST-2, RTE, and MRPC from left to right. Parameters are included in the subset according to the largest $\ell^1$ parameter norm. The random and operator norm (MLP/Attention) subsets with lowest NLL are included for comparison.

# G   CALIBRATION

In Fig. 6 the calibration curves of the MAP, Temp scaled MAP, LLLA, S-KFAC, and LA and SWAG with random module selection are shown on the SST-2 dataset. In Fig. 17 the same curves are displayed for RTE and MRPC.

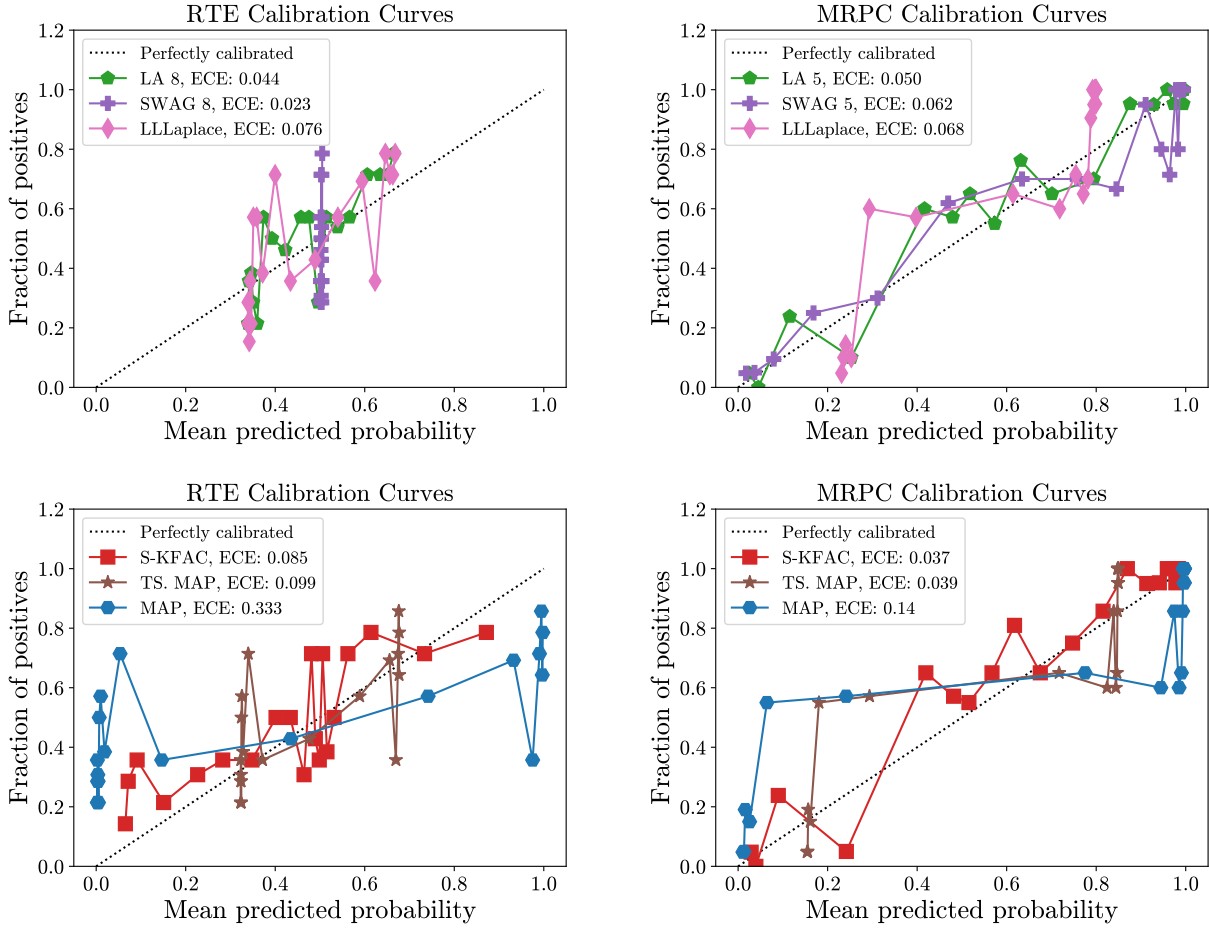

Figure 17: Calibration curves for the RTE (**left**) and MRPC (**right**) datasets shown evaluated on the test set. The median expected calibration error (ECE) across 5 runs is displayed for each method. 20 bins are placed based on the prediction probability distribution rather than uniformly. The Laplace approximation (LA) and Stochastic weight averaging - Gaussian (SWAG) are shown for **8** RTE, **4** MRPC of randomly selected stochastic neural network modules. S-KFAC is shown for $1.588\%$ of parameters being stochastic.

Fig. 17 shows S-KFAC, LA and SWAG changing the structure of posterior predictive distribution towards a more uniform distribution of mean predicted probabilities. Temperature scaling and LLLA rely on reducing the certainty of the model by "shifting" the probabilities towards 0.5.