# OpenReview forum: "Towards Scalable Bayesian Transformers: Investigating stochastic subset selection for NLP"
_auai.org/UAI/2024/Conference — UAI 2024 poster_

### Official Review · Reviewer_gKYB · 2024-03-19

**Q2-1 Originality-Novelty:** 3
**Q2-2 Correctness-Technical Quality:** 3
**Q2-5 Clarity Of Writing:** 3

**Q10 Ethical Concerns:**

no concerns

**Q1 Summary And Contributions:**

This paper studies partially stochastic Bayesian neural networks in the context of transformer models for NLP tasks for the Laplace approximation (LA) and Stochastic weight averaging - Gaussian (SWAG).

There are three contributions described in this paper:
1. Heuristics are proposed for selecting which layers to be included in the stochastic subset. They have shown that norm-based selection is promising for small subsets, and random selection is superior for larger subsets.
2. Moreover, the authors proposed Sparse-KFAC (S-KFAC), an extension of KFAC LA, which selects dense stochastic substructures of linear layers based on parameter magnitudes.
3. S-KFAC retains performance while requires a big reducing of parameters and save the usage of memory footprints.

**Q2-3 Extent To Which Claims Are Supported By Evidence:**

3: Good: the main claims are supported by convincing evidence (in the form of adequate experimental evaluation, proofs, (pseudo-)code, references, assumptions).

**Q2-4 Reproducibility:**

2: Fair: key resources (e.g. proofs, code, data) are unavailable but key details (e.g. proof sketches, experimental setup) are sufficiently well-described for an expert to confidently reproduce the main results.

**Q3 Main Strengths:**

1. The authors designed and implemented a set of numerical experiments to investigate the hypothesis of partial stochasticity improving predictive performance for transformer models in the NLP domain. This direction tends out to be

2. The authors proposed and evaluated novel heuristics for efficient identification of optimal subsets of stochastic parameters.

3. The authors demonstrated that their proposed S-KFAC Laplace approximation yields competitive predictive performance for substantially fewer stochastic parameters, leading to a significantly reduced memory footprint.

**Q4 Main Weakness:**

1. besides the comparison of the usage of memory footprint during inference, it is also important to compare the inference time, throughtput and latency as well. Prefer to learn more details in this direction.

2. subset selection is one way to reduce the memory footprint usage and if the authors' methods can be compared and discussed in a high-level of general comparison with other types of attentions such as flash-attention, of the usage of memory footprints, inference time, throughput and latency, this paper will be valued higher.

**Q5 Detailed Comments To The Authors:**

1. it is great to learn that memory footprints are drastically reduced, but how about the inference speed? could you also compare these models' inferencing speed as well? (say, throughput and latency).
2. besides this paper's selection of parameters to be used during inference, there are methods that heuristically select subsets for transformer models' inference (such as https://ojs.aaai.org/index.php/AAAI/article/view/17325). Could you also compare and discuss the major differences and the strengths of your proposals?

**Q9 Complying With Reviewing Instructions:**

Yes

---

> ### Author Rebuttal · Authors · 2024-04-08
>
> We acknowledge and thank the reviewer for investing time to evaluate our work and for the constructive feedback provided. We have tackled each comment individually in the subsequent section.
>
> ## Main Weaknesses = MW
>
> ### MW1:
> We recognize that while memory footprint might be the limiting factor for Bayesian methods applied to large language models, inference speed, latency, and throughput are also important factors for a transformer-based pipeline. The current implementation does not optimize for reduced inference time. However, it can be said that given a percentage $p$, S-KFAC will in the worst case show a $\frac{100}{p}$ speed up over a fully stochastic model. In fact distributional parameters are saved and used for calculations in matrices approximately $N_i \times N_i$ where $N_i$ is the number of stochastic parameters in layer $i$. For inference we perform several matrix-matrix multiplications, and the speedup in those is approximately cubic given a reduction in the number of stochastic parameters from the fully stochastic solution. We intend to include an empirical analysis of the S-KFAC inference time, compared to the full KFAC Laplace approximation, for our implementation and hardware.
>
> ### MW2:
> We acknowledge that it would be interesting to do a high-level comparison of the advances of Sparse-KFAC compared with other transformer-related advances. W.r.t. the flash-attention mechanism (https://arxiv.org/abs/2205.14135) a direct comparison might be misplaced. This is because flash-attention is concerned with optimizing the attention mechanism after computing $\mathbf{Q}$, $\mathbf{K}$, $\mathbf{V}$. That is $\mathbf{Q} = \mathbf{X}\mathbf{W_Q}, \quad \mathbf{K} = \mathbf{X}\mathbf{W_K}, \quad \mathbf{V} = \mathbf{X}\mathbf{W_V} \quad \mathbf{W_Q},\mathbf{W_K},\mathbf{W_V} \in \mathbf{R}^{d \times d}, \quad \mathbf{Q}, \mathbf{K}, \mathbf{V} \in \mathbf{R}^{n \times d}$. S-KFAC is concerned with adding Bayesian parameterization in $\mathbf{W_Q}$, $\mathbf{W_K}$, $\mathbf{W_V}$, rather than working on  $\mathbf{Q}, \mathbf{K}, \mathbf{V}$. With this said it would be of great interest to investigate hardware-specific optimization of S-KFAC. The block-diagonal structure inherent to the underlying KFAC methodology could facilitate a similar block-factorized optimization as is present in flash-attention.
>
> ## Detailed Comments = DC:
>
> ### DC1:
> We admit that a more comprehensive comparison of inference speed is of interest. Between the models/methods applied an interesting comparison is between posterior predictive strategies applied in SWAG and the Laplace approximation. For SWAG the approximate posterior is used to sample to approximate the posterior predictive. The current implementation uses 50 samples. Conversely, the Laplace approximation uses a linearized predictive, which is considerably less computationally demanding. Therefore in the current setting inference time using the Laplace approximation is less than for SWAG. It should however be noted that the linearized predictive is dependent on the number of outputs. Therefore for a large number of output classes, a predictive strategy based on sampling should be considered for the Laplace approximation.
>
> ### DC2:
> We thank the reviewer for the suggestion of e.g. comparing with the Informer structure. To our understanding, the paper deals with a different type of subset selection, namely with dot-product pairs in the attention mechanism, which is not a parametric operation. However, our methodology deals with the parametric components e.g. the computation of the input matrices to the attention mechanism fx. $\mathbf{Q} = \mathbf{X}\mathbf{W_Q}$. On a positive note, nothing is preventing Sparse-KFAC from being used together with the Informer architecture, as it is defined both for the convolutional and attention layers.

---

### Official Review · Reviewer_dwHQ · 2024-03-21

**Q2-1 Originality-Novelty:** 3
**Q2-2 Correctness-Technical Quality:** 3
**Q2-5 Clarity Of Writing:** 4

**Q1 Summary And Contributions:**

This work investigates approaches for creating partially stochastic Bayesian neural networks for NLP tasks. Bayesian neural networks are often not scalable, especially for models of the size used today to tackle NLP tasks. Recent works have proposed making part of a network stochastic, in order to approximate the posterior in a more compute and memory efficient manner. This work specifically looks at methods for choosing the subset of stochastic parameters in the model. It empirically evaluates these methods across several NLP tasks and using two Bayesian inference techniques, showing that their proposed technique for subnetwork selection can produce more performant models.

**Q2-3 Extent To Which Claims Are Supported By Evidence:**

4: Excellent: all claims are supported by very convincing evidence (in the form of comprehensive experimental evaluation, rigorous mathematical proofs, detailed (pseudo-)code, precise references, well-motivated and realistic assumptions) and the authors deliver what they promise.

**Q2-4 Reproducibility:**

3: Good: key resources (e.g. proofs, code, data) are available and key details (e.g. proofs, experimental setup) are sufficiently well-described for competent researchers to confidently reproduce the main results.

**Q3 Main Strengths:**

* The work addresses an important issue: making Bayesian neural networks scalable while still performant
* The prior work on which the work builds is described clearly and concisely
* The proposed subnetwork selection technique is empirically effective

**Q4 Main Weakness:**

There are a few mathematical and technical points throughout the work that need clarification, mentioned below

**Q5 Detailed Comments To The Authors:**

* It would be nice to discuss the different assumptions that SWAG/LA make and the situations for which those may or may not be suited
* In section 3, a neural network is defined as a map from R^D to R^C. The transformation of text to vector inputs is not made clear though
* Below equation 14, what is the vec(W) operation?
* An approximation to the Kronecker product is used. Are there any insights about the effects of this approximation on model validity/performance?
* I don’t understand why the fact that “derivatives are stored in terms of inputs x and output derivatives δ” causes a problem for selecting a subset of layer problems
* What is the p parameter in the first paragraph of page 5?

**Q9 Complying With Reviewing Instructions:**

Yes

---

> ### Author Rebuttal · Authors · 2024-04-08
>
> We express our thanks to the reviewer for their time and effort in evaluating our work and sharing their constructive insights. Below, you will find our point-by-point responses to each comment.
>
> ## Detailed Comments = DC
>
> ### DC1:
> We agree that a more in-depth discussion of the two methods and their assumption has relevance. One such assumption that both methods have in common is that they approximate the posterior using a Gaussian. This assumption can capture important features of the loss landscape. However real-world neural network parameter spaces are often highly complex. A Gaussian approximation can oversimplify this landscape, potentially missing critical aspects like multi-modality, skewness, or heavy tails. In practice, both methods work relatively well. Additional approximations, such as independence between entries in the model output (corresponding to different classes) is used in the Laplace approximation. This could introduce some errors when the class probabilities have high covariance.
>
> ### DC2:
> The reviewer is completely right, that we have not explained the transformation from text to vectorized input. Indeed, a significant proportion of the model's trainable parameters is found there. We will add an explanation in the camera-ready version, more informally given here: We employ the pre-trained WordPiece tokenizer from Huggingface https://huggingface.co/learn/nlp-course/en/chapter6/6. It tokenizes certain character patterns, that are hashed to a learnable vector of dimension in $R^D$. We do not make the word embeddings stochastic, and therefore only consider the subsequent part of the neural network from $R^D$ to $R^C$.
>
> ### DC3:
> We will add a sentence explaining this choice of notation in the camera-ready version. It just means to vectorize the matrix, it can be done in both a column and row major way as long as there is consistency with other vectorizations in the method.
>
> ### DC4:
> We approximate the expectation of the Kronecker product with the Kronecker product of the individual expectations following the methodology described in https://arxiv.org/abs/1503.05671. It is a major approximation to make, and the authors of that paper also realize that it likely won't hold up under any realistic assumptions. However, they argue that it captures the general structure of the GGN matrix. We are not aware of, nor have we conducted a study into the effects of this approximation, other than to say that it seems to work fairly well in practice.
>
> ### DC5:
> We will add sentences explaining this in the camera-ready version, as it is not easily seen. If one seeks to make all the parameters in the diagonal of the weight matrix stochastic i.e $W_{ii}, i = 1..N$ for some quadratic weight matrix. To perform the KFAC approximation would require storing all inputs and all output derivatives since we have one stochastic weight going from each input to each output. Hence, to decrease memory requirements, we make the dense mapping between select input and outputs stochastic instead.
>
> ### DC6:
> The parameter p denotes the percentage of inputs and outputs (corresponding to columns and rows in the weight matrix) for which we want to make the mapping between stochastic. I.e if $p=10$ we make the mapping between 10pct of the inputs and outputs stochastic.

---

### Official Review · Reviewer_P3GV · 2024-03-23

**Q2-1 Originality-Novelty:** 3
**Q2-2 Correctness-Technical Quality:** 3
**Q2-5 Clarity Of Writing:** 3

**Q1 Summary And Contributions:**

The authors provide an analysis and presentation of novel rules for selecting stochastic parameter subsets of transformers that approximate a full Bayesian parameter treatment. They demonstrate in ablation studies on the distilBERT model how various subset selection schemes perform on RMSE and calibration error metics. They show generally favorable performance of their method while reducing computational impact over previous approaches.

**Q2-3 Extent To Which Claims Are Supported By Evidence:**

2: Fair: the main claims are somewhat supported by evidence (but the experimental evaluation may be weak, or does not match entirely with the claims, important baselines may be missing, proofs contain important ideas but lack rigor, algorithmic details are only discussed superficially, references are imprecise, assumptions are not sufficiently motivated or explicated, etc.).

**Q2-4 Reproducibility:**

2: Fair: key resources (e.g. proofs, code, data) are unavailable but key details (e.g. proof sketches, experimental setup) are sufficiently well-described for an expert to confidently reproduce the main results.

**Q3 Main Strengths:**

The main strengths of the paper are both the importance of the underlying methodology (Bayesian LLM's) as well as the framework they establish for verifying and testing their methodology against baselines. For the given model architectures and datasets, experimentation is very thorough. I think that the writing and text is clear and presents a useful set of methodologies that I could see being used in practice as a practical and straightforward way to introduce uncertainty calibration to transformers without significantly modifying their architecture or underlying strengths.

**Q4 Main Weakness:**

The main practice is the lack of diversity in experimentation settings for their method. As I see it, this paper is primarily experimental in nature, as there are not theoretical results presented that justify their selection methodology against alternatives, but rather they attempt to show this experimentally. The set of experiments they perform focus on depth over breadth - ie. they only consider one architecture and 3 datasets, and I think the paper could benefit from extending to more of both the former and latter.

**Q5 Detailed Comments To The Authors:**

Limited Experimental Settings:
- I think that you have done an admirable job in evaluation your method against baselines for the settings you have chosen (distillBERT and the 3 datasets), but I think you should consider extending to more architectures and datasets, specifically those LLM architectures that tend to be used in real-world deployments today.
- I think if you repeat your same set of experiments on a few more architectures/datasets, or are able to communicate some initial results on this front for full implementation in a camera-ready, I would be willing to increase my score.
- Additionally, I think it's worth including more experimentation in terms of % of parameters that treated as a stochastic subset, beyond just the fixed 10% you use currently.

General Questions
- why the degeneracy for S-KFAC with higher module count? What is the translation from module count to percentage of total parameters? I think the latter is more informative in this regard.
- "Modular"/"module" as a term is a bit confusing to me throughout in terms of selection methodology - I think just individual weight vs matrix selection makes it more explicit (is this the correct interpretation?)
- Did you explore stochastic subsets implementation in the low-rank fine-tuning (such as LoRA) setting? Think it would be an interesting inclusion given that is the way many now interact with LLMs (rather than full pre-training).
- “Additionally, they show that inference over a stochastic subset can, at times, yield better performance than full-model stochasticity.” - any idea why?

Misc comments
- Worth introducing subnetwork inference in introduction - one or two sentences high-level overview of the idea.
- “known to exhibit pathological behavior” - I think this needs more explanation, with a definition of pathological within the context of LLM inference.
- Not clear to me why at end of section 3.0 on the dirac delta form of posterior - is \omega_S the same as \omega within the multiplication?
- Low-rank methodology could use a bit more explanation - what are the dimensionalities of the full and low-rank respectively?
- There is a bit of confusing notation throughout for Omega and subsets of Omega - not sure if referring to the full parameter set or just stochastic parameter set at certain points. I would recommend using 3 distinct subscripts of omega for full, deterministic, and stochastic parameter sets.
- “In the KFAC approximation, layers are assumed to be independent, giving rise to the block-diagonal structure” - what are the consequences of this assumption?
- Selecting only the stochastic parameters with the largest norms could potentially include all input and output neurons. → this sentence isn't clear to me as to why this would occur.
- would maybe like to see some intro theory on what portion of the full stochastic solution is recovered by your S-KFAC method, or citations to relevant works.
- Would maybe want to see more than NLL as a metric on the ramping experiments

- typo/grammar:
    - M SGD iterates → M SGD iterations.
    - Therefore, selecting a subset of the layer parameters comes with a problem. Selecting only → colon instead of period

**Q9 Complying With Reviewing Instructions:**

Yes

---

> ### Author Rebuttal · Authors · 2024-04-08
>
> We are thankful for the reviewer's commitment to examining our work and providing useful suggestions. The points raised have been addressed below.
>
> ## Limited Experimental Settings = LES
> ### LES1 \& LES2:
> We appreciate the suggestion to explore more models and concur it's a valuable future direction. Our current focus is on DistilBERT and similar smaller-scale encoders, analyzing selection strategies. While extending this to larger models like Llama variants is interesting, resource constraints limit us, as we for larger models cannot compare with 100 pct stochasticity. Our emphasis has been on assessing the impact of stochastic subset sizes and heuristics on performance across various tasks.
>
> ### LES3:
> Respectfully, we are unsure what is meant here. In Table 2 we make a method comparison, where we restrict the maximum percentage of stochastic parameters in the model to 10 percent of the total number of model parameters. In fig. 3 we make a fine-grained comparison of the percentage of stochastic parameters up to approx 30 pct. In Figure 2 we show results for S-KFAC up to approximately 12 pct stochastic parameters (corresponding approximately to p=33 from the formulae in section 4) and also compare here to 100 pct stochastic parameters.
>
> ## General Questions = GE
>
> ### GE1:
> We're unclear about the S-KFAC degeneracy reference. S-KFAC selects subsets within each network module, showing consistent performance gains with larger subsets, as seen in Fig 2, until it plateaus near or slightly below the fully stochastic model's performance. No degenerate behavior is observed at high module counts with KFAC LA approximation (refer to Figures 3 and 4).
>
> ### GE2:
> We'll clarify the module-percentage relationship in the final version. Modules, differing in size and comprising varying parameter percentages, include affine mappings within attention blocks or MLPs.
>
> ### GE3:
> We concur with exploring subset selection in low-rank fine-tuning as a valuable research direction. However, we argue that these constitute a different class of models, with possible different considerations for subset selection.
>
> ### GE4:
> We have performed our exploration of partial stochastic models for smaller-scale feed-forward models on three UCI regression datasets. We found that these conclusions could not be replicated for HMC. However, for lower-fidelity methods, such as the Laplace approximation, SWAG, and VI, we observed that partial stochasticity often outperformed full stochasticity. We hypothesize that the error in the approximation starts to outweigh the advantage of Bayesian approximation for larger percentages.
>
> ## Misc Comments = MISC
> ### MISC1:
> We agree with the reviewer that a short high-level explanation of the subnetwork inference in the introduction will improve readability. This will be included in the camera-ready version.
>
> ### MISC2:
> We thank the reviewer for the comment, and agree that pathological in terms of NLP is perhaps too harsh. For other transformer models, https://arxiv.org/abs/2110.04020 argues that VI and weight space Laplace perform poorly, or exhibit pathological behavior. We will moderate the sentence in the camera-ready version.
>
> ### MISC3, MISC5, MISC10:
> We thank you for addressing typos and unclear notations related to the Dirac delta function and $\Omega$. They will be corrected in the camera-ready version.
>
> ### MISC4:
> We are slightly confused by this comment, as it could relate to both the mention of low-rank adaption in the introduction/related works or the low-rank approximation used in the SWAG method. We will add additional details on this in the camera-ready version.
>
> ### MISC6:
> The consequence of the block diagonal structure is that we ignore the covariance structure between layers and therefore obtain a less exact approximation to the true posterior.
>
> ### MISC7:
> We agree that the mentioned sentence is confusing. We will change the sentence and elaborate further in the camera-ready version. Informally, it means that the largest weights are likely to be fairly distributed in the weight matrix and therefore span many of the columns and rows (inputs \& outputs).
>
> ### MISC8:
> We agree that S-KFAC's ability to retain a full stochastic solution could be discussed. We'll strive to elaborate further in the camera-ready paper.
>
> ### MISC9:
>  Alongside NLL, we already show ECE and will add more metrics like accuracy in the camera-ready version to provide a comprehensive performance analysis.
>
> ## Reproducibility
>
> Concerning result reproducibility, we agree with the reviewer that source code is essential to replicate the results. At the time of submission, however, the source code backing this paper was not in a state where the authors could safely say it would pass the double-blind policy. The source code will be included in the submission of the camera-ready version. The datasets used are all available online, acquiring these is automatically handled in the runner scripts in the source code.

---

### Official Review · Reviewer_e9V3 · 2024-03-24

**Q2-1 Originality-Novelty:** 2
**Q2-2 Correctness-Technical Quality:** 3
**Q2-5 Clarity Of Writing:** 2

**Q10 Ethical Concerns:**

No.

**Q1 Summary And Contributions:**

This paper investigates subnetwork inference for transformer architectures. There is extensive empirical evaluation on different subnetwork sizes and critera for selecting the subnetwork. The paper proposes a structured subnetwork strategy in the case of Laplace posterior approximation with K-FAC approximation to the Hessian.

**Q2-3 Extent To Which Claims Are Supported By Evidence:**

3: Good: the main claims are supported by convincing evidence (in the form of adequate experimental evaluation, proofs, (pseudo-)code, references, assumptions).

**Q2-4 Reproducibility:**

2: Fair: key resources (e.g. proofs, code, data) are unavailable but key details (e.g. proof sketches, experimental setup) are sufficiently well-described for an expert to confidently reproduce the main results.

**Q3 Main Strengths:**

Empirical evaluation is very extensive and convincingly demonstrates the improvement of even partial stochasticity.

**Q4 Main Weakness:**

The writing could be improved, particularly the Results section where I would encourage the authors to elaborate on the implications of their findings rather than just describing the figures and tables.

**Q5 Detailed Comments To The Authors:**

It was difficult to follow Sec. 4 (quite terse) so the presentation here could be considerably improved, especially as this section pertains to their main contribution.

**Q9 Complying With Reviewing Instructions:**

Yes

---

> ### Author Rebuttal · Authors · 2024-04-08
>
> We thank the reviewer for dedicating time to review our work and offer insightful feedback. We have responded to each specific comment in the following section.
>
> ## Main Weakness = MW
> ### MW 1:
> We agree that the writing of especially sections 4 and 5 can be difficult to follow and we will clarify these in the camera-ready version. Regarding the results section, we agree that the discussion of implications and effects of applying Sparse-KFAC are of great interest, and will be elaborated on in the camera-ready version.
>
> One implication apparent in Figure 2, is that the number of stochastic parameters needed for convergence in performance is not constant across datasets. Therefore given our current stochastic subset selection techniques, it is not possible to simply select a default percentage of stochastic parameters. Instead, when utilizing S-KFAC in a specific use case it will be advantageous to estimate the percentages p through cross-validation on a validation set, before setting a production-ready number of stochastic parameters.
>
> Another point of interest for Sparse-KFAC is its extendability to larger transformer models. The limited number of required stochastic parameters for improving performance can potentially facilitate applying Bayesian methodology to e.g. medium-sized generative models such as Mistral 7B or Llama 2 7B. The inherent modeling of uncertainty might limit the amount of hallucination produced by such models. This would however require a larger hardware setup, or applying less computationally demanding methods such as a Low-rank Adaption.
>
> Implications such as these will be discussed in the camera-ready version.
>
> ## Detailed Comments = DC
>
> ### DC 1:
> In section 4 the method of Sparse-KFAC is described, however, we realize and agree with the reviewer that the presentation is confusing. The section currently mixes analysis of the algorithm's characteristics with the introduction of the selection concept related to the Fisher block. The introduction and analysis should be split into two different subsections and possibly be supported by figures, or more coherent derivations. We will tend to this issue in the camera-ready version.
>
> ## Reproducibility
> Concerning result reproducibility, we agree with the reviewer that source code is essential to replicate the results. At the time of submission, however, the source code backing this paper was not in a state where the authors could safely say it would pass the double-blind policy. The source code will be included in the submission of the camera-ready version. The datasets used are all available online, acquiring these is automatically handled in the runner scripts in the source code.

---

### Official Review · Reviewer_cemG · 2024-03-27

**Q2-1 Originality-Novelty:** 3
**Q2-2 Correctness-Technical Quality:** 3
**Q2-5 Clarity Of Writing:** 3

**Q10 Ethical Concerns:**

No.

**Q1 Summary And Contributions:**

This work studies the problem of using subnetwork-based inference methods for uncertainty estimation of the DistilBERT model. Two methods are considered: Laplace approximation (LA) and stochastic weight averaging - Gaussian (SWAG) for comparing the selection strategies that can be either norm-based or random. A method called sparse-KFAC Laplace approximation is further proposed for using fewer parameters while still capturing important uncertainty. Empirical evaluation on three NLP datasets is presented for comparing parameter subset selection heuristics.

**Q2-3 Extent To Which Claims Are Supported By Evidence:**

3: Good: the main claims are supported by convincing evidence (in the form of adequate experimental evaluation, proofs, (pseudo-)code, references, assumptions).

**Q2-4 Reproducibility:**

3: Good: key resources (e.g. proofs, code, data) are available and key details (e.g. proofs, experimental setup) are sufficiently well-described for competent researchers to confidently reproduce the main results.

**Q3 Main Strengths:**

The study of parameter subset selection heuristics is important in the field of Bayesian deep learning for deriving scalable uncertainty estimation methods that are effective for large models.

**Q4 Main Weakness:**

- Some of the technical details are unclear to me. How is the number of largest weights of sparse-KFAC LA, min(n, m)p/100 derived? How is the hyperparameter p chosen in practice?
- While this work focuses on the DistilBERT model, it seems that there's nothing in the proposed sparse-KFAC LA specific to the transformer model and it can be applied to any neural network. Thus the empirical evaluation would be more thorough if sparse-KFAC LA is evaluated on standard benchmarks (UCI or CIFAR datasets) compared with other strong Bayesian deep learning baselines.

**Q5 Detailed Comments To The Authors:**

- Does the observations on the subset selectoin heuristics generalize to other models?

**Q9 Complying With Reviewing Instructions:**

Yes

---

> ### Author Rebuttal · Authors · 2024-04-08
>
> We appreciate the reviewer's effort to assess our work and offer valuable feedback. Below, we have addressed each comment in order.
> ## Main Weaknesses = MW
>
> ### MW 1:
> We agree that we should have written this far more clearly, and will do so in the camera-ready version. It is derived by considering a matrix  $W \in R^{n \times m}$, where $n < m$. In the S-KFAC methodology, we select the weights lying in the intersection between $\frac{n \cdot p}{100} = N$ rows and $\frac{m \cdot p}{100} = M$ columns. We select the rows and columns with the $N$  and $M$ largest $\ell^{\infty}$-norms respectively, thus making the weights in the intersection stochastic. Suppose that the largest absolute values in the matrix are all placed on different rows. Since we select $\frac{n \cdot p}{100} = N$ rows, then these will only contain the N largest absolute values, and the $N+1$'th largest value will lie on a different row, and will not be included in the stochastic subset. In the case where there are fewer columns than rows, the argument is the same if one thinks of the transposed version of the matrix. Hence we get the formula $\frac{min(n,m) \cdot p}{100}$.
>
>  We should underline how the hyperparameter is specified in practice and agree that it is an oversight on our behalf. We will add to the camera-ready version a description, more informally given here also. Determining the hyperparameter $p$ requires a similar experiment as presented in Fig. 3 and Fig. 2 where we sweep values of p in [0, 1] to find the 'near' optimal, i.e through cross-validation using the validation sets. The type of sweep can vary e.g. by choosing one's favorite algorithm e.g. bisection or GP optimization.
>
> ### MW 2:
> The reviewer is completely right that Sparse-KFAC is not limited to transformer models. We now have results showing the efficacy of Sparse-KFAC on three UCI datasets (Boston Housing, Yacht, Energy) that will be included in the camera-ready version and be compared VI, HMC, and SWAG as methods for posterior inference in the partially stochastic model. We agree with the reviewer that it would be also interesting to test the method using e.g. CNNs. However, we have not conducted these experiments yet.
>
> ## Detailed Comments = DC
> ### DC1:
> We agree with the reviewer that the generalizability of these results to other models is highly interesting. We chose the DistilBERT model to be representative of the class of BERT-like encoder models. Additionally, to the best of our knowledge, Transformer models generally deviate in terms of handling the non-parametric attention mechanism, rather than in the overall parametric structure.  We have also made results comparing Sparse-KFAC, Full Hessian Laplace, HMC, VI, and certain node-based methods for partial stochastic models in smaller feed-forward models on three UCI datasets. The results spurring the largest absolute value selection scheme in S-KFAC can also be found in Appendix B2, along with results comparing the Laplace approximation to SWAG for partially stochastic feed-forward models. We will include a more comprehensive version of these results in the camera-ready version, where we also compare the performance of S-KFAC against full Hessian Laplace and HMC to ascertain how far S-KFAC is from higher fidelity inference methods in terms of performance.

---

### Meta-Review · Area_Chair_sjXP · 2024-04-19

This paper presents strategies for selecting subsets of the parameters of a Transformer text encoder to be given approximate Bayesian treatment (with approximate inference via Laplace approx or SWAG). The paper is focused on one concrete architecture (distilBERT) as representative of BERT-type (a deep stack of Transformer-based encoder layers, pretrained for 'masked language modelling') architectures commonly used in NLP and three (binary) text classification datasets.

All reviewers agree on the relevance of the work, on the technical merits of the approach presented, and on the thoroughness of the empirical section. There was some disagreement about the clarity, with various clarification questions asked throughout, but, in my assessment, the response addressed these questions fairly well.

Additional comments:
* There was some criticism of the choice or architecture (eg, only 1 and perhaps not truly representative of the large encoders common in NLP). I'd consider demonstrating efficacy even for those larger models something like 'going the extra mile', perhaps worth of a spotlight (as a sort of reward for demonstration of impact). In any case, the potential for impact is sufficiently demonstrated in this paper.
* There was some criticism of the focus on text classification, and the authors seem to have conducted some experiments on some UCI datasets (I'd recommend having those in an appendix, even though I don't see any problem with the focus on text classification; the paper's claims are adequately scoped, in my view).
* Personally, I might have liked to see experiments beyond binary text classification, it would be great to see a small sample of the NLP landscape: some text regression (eg, machine translation quality estimation), structure prediction (eg, sequence labelling such as POS tagging, named-entity recognition, or semantic role labelling), language modelling (which you can regard as chained K-way classifiers); but then again, I'd see those as reasons to give the paper more visibility (spotlight, oral) and would not hold any of this against acceptance.

Finally, I encourage the authors to follow up on the clarification points they made and the changes they promised (most of which aimed at improving clarity and further documenting design and experimental choices), many of the questions that were asked occurred to me in my own reading of the paper.